# HSMAD: Heterophily-Driven Spectral and Manifold Learning for Graph Anomaly Detection

Chen Zhu [1]   Yaying Zhang [1]

## Abstract

Graph anomaly detection (GAD) is a fundamental task in graph learning. However, most existing methods rely on the homophily assumption, which posits that connected nodes tend to share the same labels. This assumption often fails in the presence of edge heterophily, leading to degraded performance. We first observe that down-weighting heterophilic edges, relative to the original or randomly weighted graphs, results in a more concentrated spectral energy distribution, thereby facilitating the learning of discriminative spectral representations. Moreover, existing methods typically embed graphs in Euclidean spaces, neglecting the importance of heterophily in manifold spaces. Motivated by these observations, we propose HSMAD, a novel framework for GAD. It consists of two key components: the Heterophily-Weighted Spectral Filtering module, which reconstructs the Laplacian using heterophily-based edge weighting for spectral filtering, and the Heterophily-Routed Manifold Update module, which routes neighborhood messages to the appropriate manifold for node feature updates, enabling curvature-adaptive representation learning. These spectral and geometric representations are jointly leveraged for anomaly detection. Extensive experiments on six real-world datasets show that HSMAD achieves state-of-the-art performance across the average F1-Macro, AUROC, AUPRC, and G-Mean. Specifically, the average F1-Macro score improves by 2.66% over the best-performing method.

[1]The Key Laboratory of Embedded System and Service Computing, Ministry of Education, Tongji University, Shanghai, China. Correspondence to: Yaying Zhang <yaying.zhang@tongji.edu.cn>.

*Proceedings of the 43rd International Conference on Machine Learning*, Seoul, South Korea. PMLR 306, 2026. Copyright 2026 by the author(s).

## 1. Introduction

Graph anomaly detection (GAD) (Akoglu et al., 2015; Ma et al., 2021) aims to identify nodes or edges that deviate from typical graph patterns, which is important in applications such as fraud detection (Ngai et al., 2011; Shi et al., 2022; Zhang et al., 2021), recommendation systems (Perozzi et al., 2016), and social network analysis (Yu et al., 2015; 2016). Recent advances leverage graph neural networks (GNNs) to learn node representations via neighborhood aggregation, performing well in homophilous graphs where connected nodes tend to share similar attributes or labels. However, many real-world graphs exhibit heterophily (Zhu et al., 2021; Luan et al., 2022; Zheng et al., 2026), where nodes often connect to neighbors with different attributes or labels. In such settings, aggregating dissimilar neighbors can lead to oversmoothing, which may obscure anomalies.

Heterophily affects anomaly detection along multiple dimensions (Zheng et al., 2026). Spatially, sparse anomalies can be partially overwhelmed by neighboring normal nodes. Geometrically, standard Euclidean message passing treats all neighbors uniformly, potentially ignoring structural differences between homophilous and heterophilous edges. Spectrally, heterophilous connections can disperse anomaly signals across frequencies, reducing the effectiveness of conventional spectral filters. As illustrated in Figure 1, using the Minnesota road network (Perraudin et al., 2014) with 2,642 nodes and 10% of nodes assigned as anomalies, a heterophily-weighted scheme can help concentrate spectral energy. These observations suggest that accounting for heterophily could help preserve anomaly signals across multiple domains.

Existing methods for handling heterophily primarily focus on graph structure learning, aiming to increase homophily by pruning or adding edges to better align with GNN assumptions (Gong et al., 2023; Gao et al., 2023), which may modify the original graph topology. Some methods combine spectral techniques to extract node representations from multiple frequency bands, providing complementary information across different spectral components (Wu et al., 2023; Xu et al., 2024). Manifold-based methods model the graph's intrinsic geometry to capture non-Euclidean structural properties (Gu et al., 2018; Grover et al., 2025; Dong

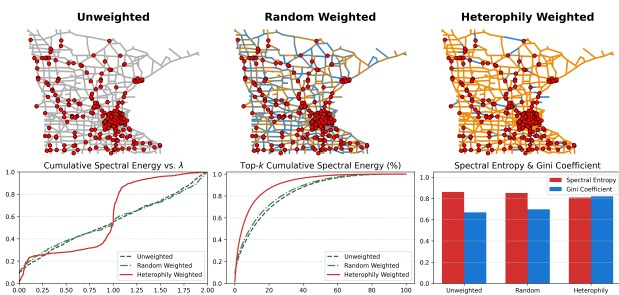

*Figure 1.* On the Minnesota road network, blue denotes low edge weight, orange high weight, and gray unweighted edges. Heterophily-weighted schemes reduce heterophilous edge weights between dissimilar nodes. The figure visualizes four spectral energy distribution metrics: (i) Cumulative energy over eigenvalues, (ii) top-$k$ cumulative energy, (iii) spectral entropy $\mathcal{H}_{\mathrm{spec}}$ and Gini coefficient $\mathcal{G}_{\mathrm{spec}}$, with lower $\mathcal{H}_{\mathrm{spec}}$ and higher $\mathcal{G}_{\mathrm{spec}}$ both indicating more concentrated energy. These metrics compare unweighted, random-weighted, and heterophily-weighted schemes' effects on anomalous nodes' spectral energy distribution; formal calculations are in section 3.3.

et al., 2025). To our knowledge, none of these methods explicitly address heterophily in the manifold space, and prior work has not jointly considered heterophily across both spectral and geometric domains. To fill this gap, we propose HSMAD, a novel framework that explicitly models heterophily in both spectral and manifold spaces. HSMAD consists of two components:

- **Heterophily-Weighted Spectral Filtering (HWSF)** first predicts edge heterophily and constructs a heterophily-weighted adjacency matrix. Edges are then selectively masked to generate two reconstructed normalized Laplacian matrices. A multi-band Beta wavelet filter bank is applied to each Laplacian, extracting node representations that capture information from both low- and high-heterophily relations, thereby enhancing the model's sensitivity to anomalous structural patterns.

- **Heterophily-Routed Manifold Update (HRMU)** leverages edge heterophily to perform weighted message passing in Euclidean space, generating neighbor aggregation messages. These messages are then routed to appropriate manifold spaces according to their heterophily: homophilic messages are mapped to spherical space, while heterophilic messages are mapped to hyperbolic space. Node representations are updated directly in the manifold spaces by integrating these curvature-aware neighbor messages with the node's own features, producing representations that adaptively capture both local homophily and heterophily.

By integrating spectral and manifold representations, HSMAD leverages information across multiple domains, enabling more effective node anomaly detection. Our main contributions can be summarized as follows:

- We introduce HSMAD, the first framework to jointly model heterophily in both spectral and geometric domains for GAD.

- We propose heterophily-weighted spectral filtering and geometry-aware manifold updates to produce more informative node representations.

- Extensive experiments on six real-world datasets demonstrate that HSMAD consistently outperforms state-of-the-art methods, highlighting its effectiveness in handling graphs with heterophily.

## 2. Related Work

**GAD Methods Addressing Heterophily.** Recent GAD methods explicitly consider heterophily to improve representation learning. Structure-based methods modify graph connectivity to reduce heterophilic interference. Specifically, GHRN (Gao et al., 2023) performs heterophily edge denoising to prune noisy inter-class edges, SparseGAD (Gong et al., 2023) sparsifies unnecessary neighbors while identifying potentially informative ones, SplitGNN (Wu et al., 2023) classifies edges to separate homophilic and heterophilic subgraphs, and NRGL (Wu et al., 2024) leverages unsupervised contrastive learning to extract and augment homophilic and heterophilic structures. Spectral-based methods exploit graph filtering to emphasize informative patterns. AMNet (Chai et al., 2022) and BWGNN (Tang et al., 2022) apply frequency-selective filters on the original graph, SEC-GFD (Xu et al., 2024) integrates hybrid low- and high-pass filtering, while LH-GNN (Wo et al., 2025) and RHO (Ai et al., 2026) design adaptive node-specific spectral filters to enhance homophily learning under varying local structures. Additionally, HUGE (Pan et al., 2025) introduces a label-free heterophily metric for unsupervised anomaly detection. Although these methods address heterophily, they primarily rely on Euclidean assumptions, neglecting the study of non-Euclidean geometric structures. This limitation may conflict with the inherent geometric properties of graphs, which are often better captured in non-Euclidean spaces.

**Special Methods.** Special GAD methods focus on specialized architectures or learning paradigms rather than explicitly modeling heterophily. PC-GNN (Liu et al., 2021) designs label-balanced node and edge sampling together with neighborhood selection to construct informative subgraphs for mini-batch training, PMP (Zhuo et al., 2024) adopts partitioned message passing with node-specific aggregation for adaptive information fusion, ConsisGAD (Chen et al., 2024) exploits unlabeled data through consistency training with learnable data augmentation, and DiG-In-GNN (Zhang

et al., 2024) introduces contrastive learning and reinforcement learning to alleviate structural inconsistency. Manifold-based methods such as SpaceGNN (Dong et al., 2025) and CurvGAD (Grover et al., 2025) learn geometry-aware representations, where SpaceGNN introduces a learnable space projection for node anomaly detection with limited labels and CurvGAD proposes a mixed-curvature graph autoencoder to characterize curvature-based geometric anomalies. DSGAD (Zheng et al., 2025) proposes dynamic wavelets with trainable filters to capture anomalous patterns, and CGADM (Wei et al., 2026) incorporates a prior-guided diffusion process for refined detection. Despite their strong empirical performance, these strategies do not explicitly model heterophily or incorporate manifold learning within a unified framework. Our HSMAD framework addresses this gap by combining spectral and manifold learning to model heterophily.

# 3. Preliminaries

## 3.1. Problem Statement

Let $G = (V, E, X)$ denote a graph, where $V$ is the set of nodes with $N = |V|$, $E$ is the set of edges, and $X \in \mathbb{R}^{N \times d}$ contains node attributes. In this paper, we focus on node-level anomaly detection, aiming to identify nodes whose attributes or connectivity patterns deviate from normal behavior. Formally, we learn a scoring function as follows:

$$s_i = f_\theta(v_i, X, A), \quad \forall v_i \in V, \tag{1}$$

where $A \in \{0, 1\}^{N \times N}$ is the adjacency matrix and $f_\theta$ is a learnable model. The score $s_i$ indicates the anomaly likelihood of node $v_i$, with larger values corresponding to higher anomaly severity.

## 3.2. Heterophily

Heterophily captures the tendency of connected nodes to differ in labels or attributes. Following prior work (Zhu et al., 2020; Pei et al., 2020; Abu-El-Haija et al., 2019), we define heterophily at both edge and node levels.

For an edge $(i, j) \in E$ connecting nodes $v_i$ and $v_j$ with labels $y_i$ and $y_j$, we define the edge heterophily as $h_{ij} = \mathbb{I}(y_i \neq y_j)$, where $\mathbb{I}(\cdot)$ is the indicator function. The graph-level edge heterophily is then computed as the average over all edges:

$$H_{\text{edge}}(G) = \frac{1}{|E|} \sum_{(i,j) \in E} h_{ij} = \frac{1}{|E|} \sum_{(i,j) \in E} \mathbb{I}(y_i \neq y_j), \tag{2}$$

where $|E|$ denotes the total number of edges in the graph.

For each $v_i \in V$, the node-level heterophily ratio is:

$$h_i = \frac{1}{|\mathcal{N}(v_i)|} \sum_{v_j \in \mathcal{N}(v_i)} \mathbb{I}(y_i \neq y_j), \tag{3}$$

where $\mathcal{N}(v_i)$ denotes the set of neighbors of $v_i$. This ratio quantifies the fraction of neighbors with different labels, characterizing local heterophily around each node.

## 3.3. Graph Laplacian and Spectral Energy

For a graph $G$, the degree matrix $D$ is diagonal with $D_{ii} = \sum_j A_{ij}$, and the symmetric normalized Laplacian (Chung, 1996) is:

$$L = I - D^{-1/2} A D^{-1/2}. \tag{4}$$

Let $\{\lambda_k, \mathbf{u}_k\}_{k=1}^N$ denote its eigenvalues and eigenvectors, ordered as $0 \leq \lambda_1 \leq \cdots \leq \lambda_N \leq 2$. Given a signal $\mathbf{x} \in \mathbb{R}^N$, its spectral projection onto the eigenvectors $\mathbf{u}_k$ is:

$$\hat{\mathbf{x}}_k = \mathbf{u}_k^\top \mathbf{x}, \quad k = 1, \ldots, N, \tag{5}$$

where $\mathbf{u}_k$ are the eigenvectors corresponding to the decomposition. The normalized spectral energy is given by:

$$\tilde{E}_k = \frac{|\hat{\mathbf{x}}_k|^2}{\sum_{j=1}^N |\hat{\mathbf{x}}_j|^2}, \quad \sum_{k=1}^N \tilde{E}_k = 1. \tag{6}$$

To characterize the distribution of spectral energy, we consider the following measures:

**Cumulative energy over eigenvalues:** The cumulative sum of $\tilde{E}_k$ ordered by the corresponding eigenvalues, which is used to assess how spectral energy is distributed across graph frequencies.

**Top-$k$ cumulative energy:** The sum of the $k$ largest $\tilde{E}_k$ values, which indicates the concentration of energy in the dominant modes.

**Spectral entropy:** Defined as $\mathcal{H}_{\text{spec}} = -\sum_{k=1}^N \tilde{E}_k \log \tilde{E}_k$, where lower values of $\mathcal{H}_{\text{spec}}$ indicate a more concentrated energy distribution.

**Gini coefficient:** Given by $\mathcal{G}_{\text{spec}} = \frac{\sum_{i,j} |\tilde{E}_i - \tilde{E}_j|}{2N \sum_{k=1}^N \tilde{E}_k}$, where higher values of $\mathcal{G}_{\text{spec}}$ reflect a stronger concentration of energy in fewer modes.

## 3.4. $\kappa$-Stereographic Model

We briefly review the stereographic model of constant-curvature manifolds following the unified $\kappa$-geometry framework (Bachmann et al., 2020; Ganea et al., 2018). Let $\mathcal{M}_\kappa^d$ denote a $d$-dimensional Riemannian manifold with constant sectional curvature $\kappa \in \mathbb{R}$, where $\kappa < 0$, $\kappa = 0$, and $\kappa > 0$ correspond to hyperbolic, Euclidean, and spherical geometries, respectively.

For $x \in \mathbb{R}$, we define the curvature-dependent trigonometric functions $\tan_\kappa$ and its inverse $\tan_\kappa^{-1}$ in a unified piecewise manner:

$$f_\kappa(y) = \begin{cases} \frac{1}{\sqrt{-\kappa}} \phi(\sqrt{-\kappa}\, y), & \kappa < 0, \\ y, & \kappa = 0, \\ \frac{1}{\sqrt{\kappa}} \phi(\sqrt{\kappa}\, y), & \kappa > 0, \end{cases} \quad (7)$$

where for the forward function $\tan_\kappa$, $\phi = \tanh$ for $\kappa < 0$ and $\phi = \tan$ for $\kappa > 0$, and for the inverse $\tan_\kappa^{-1}$, $\phi$ is replaced by $\operatorname{arctanh}$ or $\arctan$ respectively.

For two points $\mathbf{x}, \mathbf{y} \in \mathcal{M}_\kappa^d$ represented in the stereographic model, the $\kappa$-addition is defined as:

$$\mathbf{x} \oplus_\kappa \mathbf{y} = \frac{(1 - 2\kappa\langle\mathbf{x},\mathbf{y}\rangle - \kappa\|\mathbf{y}\|^2)\mathbf{x} + (1 + \kappa\|\mathbf{x}\|^2)\mathbf{y}}{1 - 2\kappa\langle\mathbf{x},\mathbf{y}\rangle + \kappa^2\|\mathbf{x}\|^2\|\mathbf{y}\|^2}. \quad (8)$$

When $\kappa < 0$, this operation reduces to the standard Möbius addition in the Poincaré ball, and when $\kappa = 0$ it degenerates to Euclidean vector addition.

Let $\mathbf{x} \in \mathcal{M}_\kappa^d$ be a base point and $\mathbf{v} \in T_\mathbf{x}\mathcal{M}_\kappa^d$ a tangent vector. Define the conformal factor $\lambda_\kappa^\mathbf{x} = 2/(1 + \kappa\|\mathbf{x}\|^2)$. The exponential map at $\mathbf{x}$ is given by:

$$\exp_\kappa^\mathbf{x}(\mathbf{v}) = \mathbf{x} \oplus_\kappa \left( \tan_\kappa\left( \frac{\lambda_\kappa^\mathbf{x}}{2}\|\mathbf{v}\| \right) \frac{\mathbf{v}}{\|\mathbf{v}\|} \right), \quad (9)$$

and the logarithmic map at $\mathbf{x}$ is:

$$\log_\kappa^\mathbf{x}(\mathbf{y}) = \frac{2}{\lambda_\kappa^\mathbf{x}} \tan_\kappa^{-1}(\|(-\mathbf{x}) \oplus_\kappa \mathbf{y}\|) \frac{(-\mathbf{x}) \oplus_\kappa \mathbf{y}}{\|(-\mathbf{x}) \oplus_\kappa \mathbf{y}\|}. \quad (10)$$

In this work, both exponential and logarithmic maps use the origin as the base point throughout the propagation layers. This simplifies computation and provides a shared reference space for message aggregation.

# 4. Methodology

As illustrated in Figure 2, we first present an overview of the proposed HSMAD framework. Section 4.1 describes Heterophily-Weighted Spectral Filtering, which extracts heterophily-weighted spectral representations. Section 4.2 introduces Heterophily-Routed Manifold Update, which performs representation updating across different manifold channels. Section 4.3 then presents the final anomaly detection module, which computes anomaly scores from the learned representations.

## 4.1. Heterophily-Weighted Spectral Filtering

We first predict edge heterophily scores to construct weighted Laplacian matrices. These matrices, along with node attributes, are then processed through a multi-scale filter bank to generate spectral node representations.

**Edge Heterophily Prediction.** Given a graph $G$ with node attributes $\mathbf{X} \in \mathbb{R}^{N \times d}$, where $\mathbf{x}_i \in \mathbb{R}^d$ denotes the raw attributes of node $i$, we construct the feature for each edge $(i, j) \in \mathcal{E}$ as the absolute difference of the incident node attributes:

$$\mathbf{x}_{ij} = |\mathbf{x}_i - \mathbf{x}_j| \in \mathbb{R}^d. \quad (11)$$

To capture heterophily patterns in the graph, we employ a trainable edge predictor $g_\phi$, implemented as a multi-layer perceptron (MLP), to estimate the heterophily score for each edge $(i, j)$:

$$\hat{e}_{ij} = \sigma(g_\phi(\mathbf{x}_{ij})) \in [0, 1], \quad (12)$$

where $\sigma(\cdot)$ is the sigmoid function.

The predictor is trained using the binary cross-entropy loss:

$$\mathcal{L}_{\text{edge}} = - \sum_{(i,j) \in \mathcal{E}} \left[ y_{ij} \log \hat{e}_{ij} + (1 - y_{ij}) \log(1 - \hat{e}_{ij}) \right], \quad (13)$$

where $y_{ij} \in \{0, 1\}$ indicates the ground-truth heterophily of the edge, with $y_{ij} = 1$ for heterophilous edges and $y_{ij} = 0$ for homophilous edges.

**Weighted Laplacian Construction.** We first assign edge weights $w_{ij} = \exp(-\alpha_w \hat{e}_{ij})$ with $\alpha_w > 0$, and construct the weighted adjacency matrix $A_w$. The corresponding normalized Laplacian $L_w$ is defined in the standard way.

To separate edges by heterophily, we introduce a quantile threshold $\tau$ and define binary masks $\mathcal{M}^{\text{low}}$ and $\mathcal{M}^{\text{high}}$. The masked adjacency matrices are obtained by element-wise multiplication $A_w^m = \mathcal{M}^m \odot A_w$, for $m \in \{\text{low}, \text{high}\}$, and the corresponding normalized Laplacians $L_w^m$ are computed accordingly. Each $L_w^m$ is symmetric, positive semi-definite, and its eigenvalues lie in $[0, 2]$ (see Lemma 4.1). The proof is provided in Appendix A.

**Lemma 4.1** (Spectral Bounds under Heterophily Weighting). *Let $A_w$ be the weighted adjacency matrix of an undirected graph with symmetric edge weights $w_{uv} = w_{vu} \in [0, 1]$. Then the normalized Laplacian $L_w$ is symmetric and positive semi-definite, and its spectrum satisfies $\Lambda(L_w) \subset [0, 2]$.*

**Heterophily-Weighted Filter Bank.** We adopt the Beta distribution as a flexible spectral kernel for graphs, following BWGNN (Tang et al., 2022). Defined on the interval $w \in [0, 1]$, the Beta distribution considered here is parameterized by two non-negative integer parameters $p, q$ and has the probability density:

$$\beta_{p,q}(w) = \frac{1}{B(p+1, q+1)} w^p (1-w)^q, \quad (14)$$

where $B(p+1, q+1)$ is the Beta function. Since $p$ and $q$ are non-negative integers, it can be computed as:

$$B(p+1, q+1) = \frac{p!\, q!}{(p+q+1)!}. \quad (15)$$

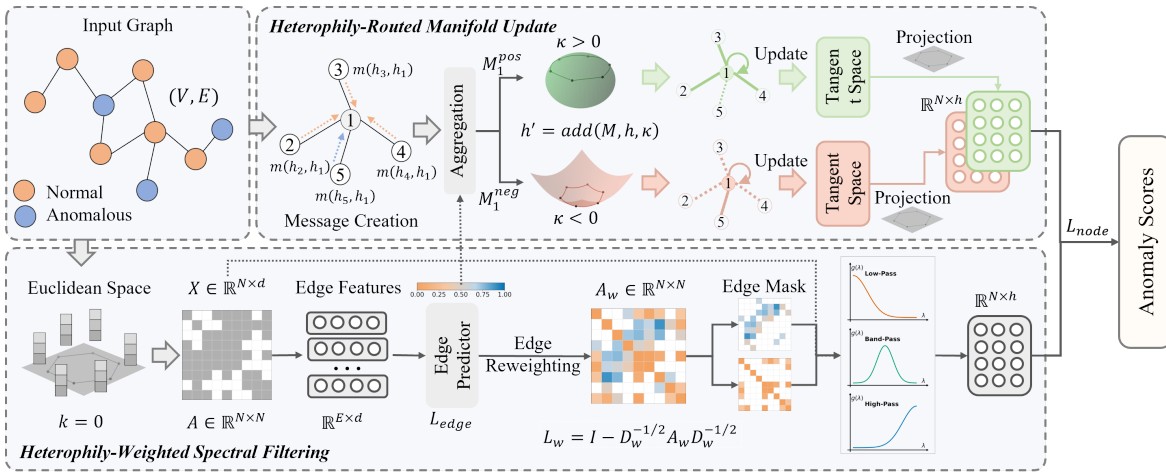

*Figure 2.* Overview of the proposed HSMAD. It comprises two modules for heterophily modeling: (1) HWSF predicts edge heterophily, constructs heterophily-weighted Laplacians, and uses a multi-band Beta wavelet filter bank to generate low- and high-heterophily spectral representations; (2) HRMU leverages curvature-aware learning on two manifolds for message routing, emphasizing homophilic/heterophilic messages in spherical/hyperbolic spaces to update node representations.

By fixing $p + q = C$, where $C$ is a non-negative integer, we construct a multi-scale Beta wavelet filter bank on $L_w$. The filter bank contains $C + 1$ filters, where the $k$-th filter is defined as a Beta wavelet with parameters $(k, C - k)$ applied to $L_w$:

$$W_k = \beta^*_{k,C-k}(L_w), \qquad (16)$$

where the scaled Beta polynomial is defined as follows:

$$\beta^*_{k,C-k}(\lambda) = \frac{(\lambda/2)^k (1 - \lambda/2)^{C-k}}{2B(k+1, C-k+1)}. \qquad (17)$$

In this filter bank, $W_0$ is a low-pass filter, $W_C$ is a high-pass filter, and intermediate $W_k$ ($0 < k < C$) are band-pass filters. The $\lambda/2$ scaling maps the Beta polynomial to the Laplacian spectrum $[0, 2]$, ensuring the filters are well-defined and efficiently computable via polynomial approximation.

**Lemma 4.2** (Beta Graph Wavelet Frame). *Let $L_w$ be a symmetric matrix with spectrum $\Lambda(L_w) \subset [0, 2]$, and let $\mathcal{W} = \{W_0, W_1, \ldots, W_C\}$ be the multi-scale Beta filter bank applied to $L_w$. Then $\mathcal{W}$ forms a stable graph wavelet frame: there exist constants $0 < A \leq B < \infty$ such that:*

$$A\|x\|_2^2 \leq \sum_{k=0}^{C} \|W_k x\|_2^2 \leq B\|x\|_2^2, \quad \forall x \in \mathbb{R}^N. \qquad (18)$$

Lemma 4.2 ensures that the multi-scale Beta wavelet filter bank forms a stable graph wavelet frame, guaranteeing stable spectral representations. The proof is provided in Appendix A.

**Spectral Representations.** Before spectral filtering, original node attributes are first mapped into an $h$-dimensional hidden space:

$$\mathbf{H}_0 = \mathrm{MLP}(\mathbf{X}) \in \mathbb{R}^{N \times h}. \qquad (19)$$

We then apply the multi-scale Beta wavelet filter bank to generate spectral representations from the weighted Laplacians. Let $L_w^m$ denote the weighted Laplacian for branch $m \in \{\text{low}, \text{high}\}$, the multi-scale node representations can be computed as:

$$\mathbf{H}^m = \left[ W_k^m \mathbf{H}_0 \right]_{k=0}^{C}, \qquad (20)$$

where $[\cdot]_{k=0}^{C}$ denotes feature-wise concatenation over all filter orders.

The two filtered representations are then fused through a learnable gating mechanism and further projected to obtain the final spectral node representations $\mathbf{H}^s \in \mathbb{R}^{N \times h}$. These representations integrate information from both low- and high-heterophily edges. The detailed procedure is summarized in Algorithm 1 in Appendix D.

### 4.2. Heterophily-Routed Manifold Update

We first provide a detailed explanation of the heterophily-routed message creation process, followed by a description of how the updated node features are fused to generate manifold representations.

**Feature Update Mechanism.** Let $\mathbf{h}_i^{(l)} \in \mathbb{R}^h$ denote the representation of node $i$ at layer $l$. Node features are linearly transformed and optionally normalized by node degrees:

$$\tilde{\mathbf{h}}_j^{(l)} = \frac{1}{\sqrt{d_j}} W^\top \mathbf{h}_j^{(l)}, \qquad \tilde{\mathbf{h}}_i^{(l)} = W_0^\top \mathbf{h}_i^{(l)}, \qquad (21)$$

where $W$ and $W_0$ are learnable weight matrices. Neighbor features are aggregated as a weighted sum:

$$\mathbf{m}_i^{(l)} = \sum_{j \in \mathcal{N}(i)} w_{ji} \tilde{\mathbf{h}}_j^{(l)}. \tag{22}$$

Both the aggregated message and the node's own transformed feature are mapped to the curvature manifold via the exponential map:

$$\hat{\mathbf{m}}_i^{(l)} = \exp_\kappa^{\mathbf{0}}(\mathbf{m}_i^{(l)}), \quad \hat{\mathbf{h}}_i^{(l)} = \exp_\kappa^{\mathbf{0}}(\tilde{\mathbf{h}}_i^{(l)}), \tag{23}$$

and combined using curvature-aware addition, followed by a logarithmic map back to the Euclidean tangent space:

$$\mathbf{h}_i^{(l+1)} = \log_\kappa^{\mathbf{0}} \left( \hat{\mathbf{m}}_i^{(l)} \oplus_\kappa \hat{\mathbf{h}}_i^{(l)} \right), \tag{24}$$

where the manifold operations $\exp_\kappa^{\mathbf{0}}$, $\log_\kappa^{\mathbf{0}}$, and $\oplus_\kappa$ are defined in Section 3.4.

To handle heterophilic edges, we extend the feature update by incorporating two curvature channels. Each edge $(j, i)$ is assigned a predicted heterophily score $\hat{e}_{ji}$, which determines the routing channel it follows via:

$$w_{ji}^+ = \exp(-\alpha \hat{e}_{ji}), \tag{25}$$

$$w_{ji}^- = 1 - \exp(-\alpha \hat{e}_{ji}), \tag{26}$$

where $\alpha > 0$ is learnable. Messages from primarily homophilic edges (low $\hat{e}_{ji}$) are routed to the positive-curvature channel, while messages from highly heterophilic edges (high $\hat{e}_{ji}$) are routed to the negative-curvature channel.

Within each channel, the node feature $\tilde{\mathbf{h}}_i^{(l)}$ and the aggregated message $\mathbf{m}_i^{(l,\pm)}$ are combined via $\kappa$-addition:

$$\mathbf{h}_i^{(l+1,\pm)} = \log_{\kappa^\pm}^{\mathbf{0}} \left( \exp_{\kappa^\pm}^{\mathbf{0}}(\tilde{\mathbf{h}}_i^{(l)}) \oplus_{\kappa^\pm} \exp_{\kappa^\pm}^{\mathbf{0}}(\mathbf{m}_i^{(l,\pm)}) \right). \tag{27}$$

The curvature parameters $\kappa^+$ and $\kappa^-$ are learnable. They are initialized near zero and passed through a tanh function to produce bounded values, after which positive and negative signs are assigned to $\kappa^+$ and $\kappa^-$, respectively.

**Lemma 4.3** (First-Order Curvature Expansion of $\kappa$-Addition). *Let $\mathbf{x}, \mathbf{y} \in \mathbb{R}^h$ be vectors near the origin, and let $\oplus_\kappa$ denote the stereographic $\kappa$-addition in a constant-curvature manifold of curvature $\kappa$. Then, at first order, the manifold sum is the usual Euclidean sum $\mathbf{x} + \mathbf{y}$, plus a curvature-induced correction $\kappa(\|\mathbf{x}\|^2\mathbf{y} - \|\mathbf{y}\|^2\mathbf{x}) + 2\kappa\langle\mathbf{x}, \mathbf{y}\rangle\mathbf{y} + R_\kappa(\mathbf{x}, \mathbf{y})$, where the first term encodes a directional deviation due to curvature and the second term is a colinear scaling along $\mathbf{y}$. In particular, $\mathbf{x} \oplus_\kappa \mathbf{y} \neq \mathbf{y} \oplus_\kappa \mathbf{x}$ in general for $\kappa \neq 0$.*

Lemma 4.3 gives a first-order curvature expansion of $\kappa$-addition, with the proof provided in Appendix A. As

$\kappa^\pm \to 0$, the curvature-dependent terms vanish and the intermediate $\kappa$-addition reduces to Euclidean addition. The positive and negative channels apply this operation with opposite curvature signs before their resulting representations are passed to the fusion mechanism.

**Manifold Representations.** For each node $i$, the positive- and negative-curvature representations are fused using a learnable gate:

$$\mathbf{h}_i^{(l+1)} = \gamma \, \mathbf{h}_i^{(l+1,+)} + (1-\gamma) \, \mathbf{h}_i^{(l+1,-)}, \tag{28}$$

where $\gamma$ is the gating coefficient. At the matrix level, this produces the final manifold node representations $\mathbf{H}^c \in \mathbb{R}^{N \times h}$, with each row corresponding to a curvature-aware node representation. The detailed procedure is summarized in Algorithm 2 in Appendix D.

### 4.3. Anomaly Detection

Anomalous nodes are detected by combining spectral and manifold representations. For each node $v_i \in V$, the two representations are fused via a learnable gate to form the final node representation $\mathbf{h}_i^{\text{final}}$. The predicted probability of node $v_i$ being anomalous is computed as:

$$\hat{y}_i = \text{sigmoid}\left((W_{\text{out}} \, \mathbf{h}_i^{\text{final}})\right), \tag{29}$$

where $W_{\text{out}}$ is a learnable linear projection.

The node-level loss is defined as binary cross-entropy over labeled training nodes:

$$\mathcal{L}_{\text{node}} = - \sum_{i \in \mathcal{V}_{\text{train}}} \left[ y_i \log \hat{y}_i + (1 - y_i) \log(1 - \hat{y}_i) \right], \tag{30}$$

where $y_i \in \{0, 1\}$ is the ground-truth label.

The total loss jointly optimizes node classification and edge heterophily prediction:

$$\mathcal{L}_{\text{total}} = w_{\text{node}} \, \mathcal{L}_{\text{node}} + w_{\text{edge}} \, \mathcal{L}_{\text{edge}}, \tag{31}$$

where $\mathcal{L}_{\text{edge}}$ is defined in Section 4.1, and $w_{\text{node}}, w_{\text{edge}}$ are dynamically adjusted during training based on their relative magnitudes, with a small constant added to ensure numerical stability. Each weight is computed to balance the contribution of node and edge losses.

## 5. Experiments

In this section, we systematically present the experimental setup, datasets, baselines, and performance comparison. We then report the results of ablation study and sensitivity analysis, followed by a case study.

*Table 1.* Performance comparison of all datasets in terms of F1-Macro and AUROC. Bold and underlined values indicate the best and second-best results, respectively. "OOM" denotes out of memory, and "OOT" denotes out of time (running longer than one day).

| Method | Amazon | | Yelp | | Weibo | | Tolokers | | T-Finance | | T-Social | | Avg. | |
|---|---|---|---|---|---|---|---|---|---|---|---|---|---|---|
| | F1-Macro | AUROC | F1-Macro | AUROC | F1-Macro | AUROC | F1-Macro | AUROC | F1-Macro | AUROC | F1-Macro | AUROC | F1-Macro | AUROC |
| MLP | 0.9223 | 0.9801 | 0.6885 | 0.8202 | 0.9009 | 0.9119 | 0.6169 | 0.7389 | 0.8458 | 0.9114 | 0.5450 | 0.6706 | 0.7532 | 0.8389 |
| GCN | 0.6396 | 0.8004 | 0.5635 | 0.5988 | 0.9502 | 0.9791 | 0.6329 | 0.7561 | 0.8442 | 0.9154 | 0.6856 | 0.7011 | 0.7193 | 0.7918 |
| GAT | 0.9216 | 0.9756 | 0.6736 | 0.8037 | 0.9408 | 0.9614 | 0.6573 | 0.7877 | 0.7860 | 0.9234 | 0.6073 | 0.7163 | 0.7644 | 0.8614 |
| GraphSAGE | 0.8245 | 0.9110 | 0.7159 | 0.8495 | 0.8984 | 0.9320 | 0.6580 | 0.7864 | 0.6855 | 0.7594 | 0.5804 | 0.6074 | 0.7271 | 0.8076 |
| AMNet | 0.9083 | 0.9552 | 0.6807 | 0.8385 | 0.9098 | 0.9424 | 0.4433 | 0.6745 | 0.8325 | 0.9224 | 0.4793 | 0.3295 | 0.7090 | 0.7771 |
| BWGNN | 0.9196 | 0.9649 | 0.7360 | 0.8595 | 0.9302 | 0.9722 | 0.6712 | 0.7993 | 0.9084 | 0.9600 | 0.8811 | 0.9772 | 0.8411 | 0.9222 |
| GHRN | 0.9207 | 0.9741 | 0.7286 | 0.8587 | 0.9150 | 0.9666 | 0.6597 | 0.7898 | 0.8618 | 0.9393 | 0.9087 | 0.9853 | 0.8324 | 0.9190 |
| SparseGAD | 0.9013 | 0.9714 | 0.7025 | 0.8520 | 0.9305 | 0.9420 | 0.5183 | 0.7623 | 0.8747 | 0.9403 | OOM | OOM | 0.7855 | 0.8936 |
| SEC-GFD | 0.9188 | 0.9786 | 0.7369 | 0.8608 | 0.9297 | 0.9725 | 0.6729 | 0.7962 | 0.9001 | 0.9568 | 0.8670 | 0.9653 | 0.8376 | 0.9217 |
| NRGL | 0.9027 | 0.9275 | 0.6503 | 0.7613 | 0.7925 | 0.8294 | 0.6275 | 0.7368 | 0.8107 | 0.9070 | OOM | OOM | 0.7567 | 0.8324 |
| PC-GNN | 0.8902 | 0.9453 | 0.6465 | 0.7697 | 0.8533 | 0.9181 | 0.6080 | 0.7272 | 0.8415 | 0.9322 | 0.5132 | 0.6698 | 0.7255 | 0.8271 |
| ConsisGAD | 0.9156 | 0.9786 | 0.6997 | 0.8332 | 0.9164 | 0.9669 | 0.6447 | 0.7724 | 0.9134 | 0.9721 | 0.8148 | 0.9618 | 0.8174 | 0.9142 |
| PMP | 0.9177 | 0.9757 | 0.8192 | 0.9382 | 0.9396 | 0.9810 | 0.6523 | 0.8144 | 0.9163 | 0.9689 | 0.9433 | 0.9946 | 0.8647 | 0.9455 |
| DSGAD | 0.9213 | 0.9775 | 0.7253 | 0.8496 | 0.9406 | 0.9776 | 0.6782 | 0.8023 | 0.9217 | 0.9628 | 0.9106 | 0.9801 | 0.8496 | 0.9250 |
| CurvGAD | 0.9232 | 0.9572 | 0.6330 | 0.7403 | 0.8746 | 0.8898 | 0.6200 | 0.7401 | 0.9062 | 0.9529 | OOM | OOM | 0.7914 | 0.8561 |
| SpaceGNN | 0.9247 | 0.9693 | 0.7803 | 0.8959 | 0.9476 | 0.9887 | 0.6527 | 0.7888 | 0.9258 | 0.9675 | OOT | OOT | 0.8462 | 0.9220 |
| CGADM | 0.9184 | 0.9774 | 0.8408 | 0.9428 | 0.9529 | 0.9925 | 0.6592 | 0.7810 | 0.9020 | 0.9113 | 0.9591 | 0.9932 | 0.8721 | 0.9330 |
| HSMAD | 0.9283 | 0.9827 | 0.8682 | 0.9447 | 0.9554 | 0.9944 | 0.7275 | 0.8439 | 0.9290 | 0.9751 | 0.9634 | 0.9888 | 0.8953 | 0.9549 |

*Table 2.* Ablation study on six datasets in terms of F1-Macro and AUROC. "w/o" denotes removing the corresponding component.

| Method | Amazon | | Yelp | | Weibo | | Tolokers | | T-Finance | | T-Social | | Avg. | |
|---|---|---|---|---|---|---|---|---|---|---|---|---|---|---|
| | F1-Macro | AUROC | F1-Macro | AUROC | F1-Macro | AUROC | F1-Macro | AUROC | F1-Macro | AUROC | F1-Macro | AUROC | F1-Macro | AUROC |
| w/o HWSF | 0.9234 | 0.9812 | 0.8306 | 0.9295 | 0.9548 | 0.9931 | 0.7206 | 0.8376 | 0.9170 | 0.9709 | 0.9392 | 0.9766 | 0.8809 | 0.9482 |
| w/o HRMU | 0.9269 | 0.9796 | 0.8416 | 0.9306 | 0.9535 | 0.9900 | 0.7114 | 0.8373 | 0.9220 | 0.9668 | 0.9249 | 0.9692 | 0.8801 | 0.9456 |
| HSMAD | 0.9283 | 0.9827 | 0.8682 | 0.9447 | 0.9554 | 0.9944 | 0.7275 | 0.8439 | 0.9290 | 0.9751 | 0.9634 | 0.9888 | 0.8953 | 0.9549 |

## 5.1. Experimental Setup

All experiments are conducted on a server with an NVIDIA Tesla V100 GPU and 32 GB of memory, except for the large-scale T-Social dataset, which is trained on an NVIDIA DGX Spark machine with 128 GB of memory. To ensure a fair comparison, each method is run ten times with different random seeds, and the average results are reported. Each dataset is randomly split into 40% training, 20% validation, and 40% testing. The methods are evaluated using F1-Macro, AUROC, AUPRC, and G-Mean, with detailed definitions and calculation formulas provided in Appendix C. All models are optimized using the Adam optimizer, with a hidden dimension of 64 for all datasets except T-Social, where it is set to 10, and a fixed learning rate of 0.01. For the baseline methods, we adopt the recommended hyperparameter settings from their official open-source implementations. The code for this work is available at https://github.com/cozy24/HSMAD.

## 5.2. Datasets and Baselines

We conduct experiments on six widely adopted real-world datasets for GAD, including Amazon, Yelp, Weibo, Tolokers, T-Finance, and T-Social (Tang et al., 2023), all of which are commonly used for anomaly detection tasks. Comprehensive dataset statistics are provided in Appendix B. Among these, T-Social is a large-scale dataset, containing 5,781,065 nodes and 73,105,508 edges, while Weibo is a smaller dataset, and Amazon, Yelp, Tolokers, and T-Finance are of medium scale.

We compare HSMAD with both classical and recent methods. The classical methods include MLP (Rosenblatt, 1958), GCN (Kipf, 2017), GraphSAGE (Hamilton et al., 2017), and GAT (Veličković et al., 2018). Among recent GAD methods, we consider two categories. The first category consists of heterophily-aware methods that explicitly address heterophilic structures, including AMNet (Chai et al., 2022), BWGNN (Tang et al., 2022), SparseGAD (Gong et al., 2023), GHRN (Gao et al., 2023), SEC-GFD (Xu et al., 2024), and NRGL (Wu et al., 2024). The second category includes specialized GAD methods based on distinctive architectures or learning paradigms, including PC-GNN (Liu et al., 2021), ConsisGAD (Chen et al., 2024), PMP (Zhuo et al., 2024), DSGAD (Zheng et al., 2025), CurvGAD (Grover et al., 2025), SpaceGNN (Dong et al., 2025), and CGADM (Wei et al., 2026).

## 5.3. Performance Comparison

Table 1 summarizes the performance of all methods on six real-world datasets, evaluated using F1-Macro and AUROC. HSMAD achieves state-of-the-art performance in terms of F1-Macro, outperforming all competing methods across all datasets. In terms of AUROC, HSMAD leads on five of the six datasets, surpassing all other methods. However, on the T-Social dataset, HSMAD slightly lags behind PMP and

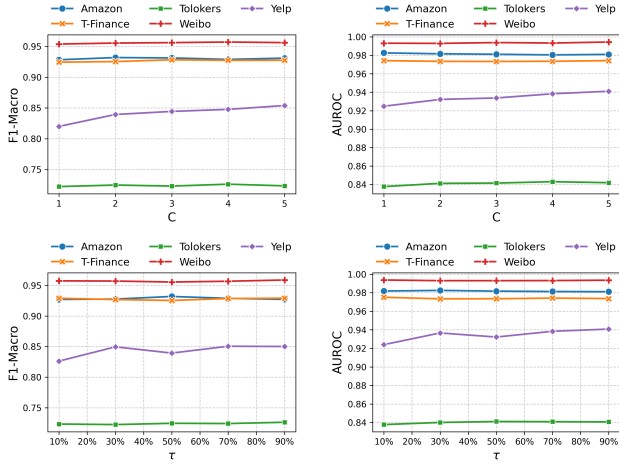

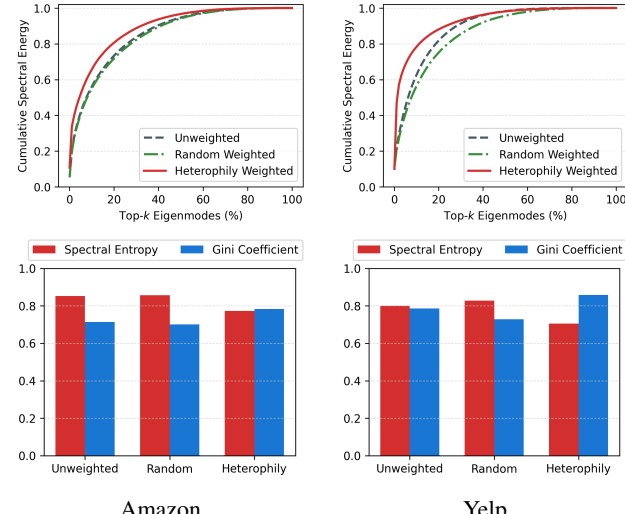

*Figure 3.* Sensitivity analysis of HSMAD with respect to the spectral filter order $C$ and the quantile $\tau$. Top row: F1-Macro and AUROC on Amazon, Yelp, Weibo, Tolokers, and T-Finance with fixed $\tau = 0.5$ and varying $C$. Bottom row: F1-Macro and AUROC on the same datasets with fixed $C = 2$ and varying $\tau$.

*Figure 4.* Effects of unweighted, random-weighted, and heterophily-weighted schemes on the top-$k$ cumulative energy distribution, spectral entropy, and Gini coefficient.

CGADM. The T-Social dataset is characterized by a relatively low average node degree (see Appendix B), limiting the availability of informative neighborhood signals. Since HSMAD relies on edge heterophily for spectral filtering and manifold learning, the sparse connectivity in T-Social likely hinders the effectiveness of heterophily-based modeling, contributing to the observed performance gap.

Additionally, HSMAD obtains the highest average F1-Macro and AUROC across all datasets, which further confirms its state-of-the-art performance under both threshold-dependent and ranking-based evaluation metrics. Supplementary comparisons using AUPRC and G-Mean are provided in Appendix E. The consistent advantages observed across different metrics further demonstrate the robustness of HSMAD under diverse evaluation criteria.

### 5.4. Ablation Study

We conducted a set of ablation experiments to evaluate the contribution of the main components in HSMAD. Specifically, we consider two variants by removing the Heterophily-Weighted Spectral Filtering module (w/o HWSF) and the Heterophily-Routed Manifold Update module (w/o HRMU), respectively. Table 2 reports the comparison results on six datasets using F1-Macro and AUROC as the evaluation metrics. Additional ablation results under AUPRC and G-Mean are provided in Appendix F.

The results show that removing either HWSF or HRMU leads to consistent performance degradation across all datasets. This indicates that both modules make meaningful contributions to the overall effectiveness of HSMAD. It is also worth noting that, even after removing one of the two

components, the resulting HSMAD variants still maintain competitive performance compared with existing baselines, which demonstrates the robustness of the overall framework.

### 5.5. Sensitivity Analysis

We examine the impact of two key hyperparameters: spectral filter order $C$ and edge mask quantile threshold $\tau$. Figure 3 shows the sensitivity across five datasets. The detailed sensitivity analysis for all five datasets are provided in Appendix G.

For Yelp, the detection performance is more sensitive to both hyperparameters. As $C$ increases, both F1-Macro and AUROC improve, highlighting the advantage of capturing higher-order spectral patterns. The quantile threshold $\tau$, which represents the heterophily score quantile, also influences performance. As $\tau$ increases from 10% to 30%, performance improves, then slightly drops, before rising again at higher values. This indicates that Yelp's performance is particularly sensitive to edge masking. In contrast, the other datasets exhibit more stable performance, suggesting that moderate values of $C$ and typical quantile thresholds are sufficient for effective representation learning.

### 5.6. Case Study

We conducted a case study on Amazon and Yelp subgraphs, each with 2,000 nodes, where anomalies are represented by a binary indicator signal derived from node labels. We performed Laplacian spectral decomposition and projected the anomaly signal onto the eigenvectors to obtain the spectral energy distribution. Figure 4 shows the average normalized cumulative energy of the top-$k$ eigenvectors, along

with spectral entropy and Gini coefficient, for unweighted, random-weighted, and heterophily-weighted schemes. The edge weights for the weighted schemes are obtained from the model during training. Faster-growing top-$k$ curves, lower entropy, and higher Gini values indicate that the anomaly signal's energy is concentrated in fewer eigen-modes. Across both datasets, the heterophily-weighted scheme consistently demonstrates this concentration, highlighting its effectiveness in focusing the anomaly signal's energy.

## 6. Conclusion

In this paper, we propose HSMAD, a framework for GAD that explicitly models heterophily in both spectral and manifold domains. HSMAD includes a spectral filtering module that reconstructs spectral features via heterophily-weighted edges, and a manifold update module that updates representations through heterophily-routed message propagation. Experiments on six real-world datasets demonstrate consistent improvements. Future work includes reducing computational and memory overhead and extending HSMAD to dynamic or heterogeneous graphs.

## Acknowledgements

This work was partly supported by the National Key Research and Development Program of China under Grant 2023YFB3002201, the National Natural Science Foundation of China under Grant 72342026, and Fundamental Research Funds for the Central Universities under Grant 2024-6-ZD-02.

## Impact Statement

This paper presents work whose goal is to advance the field of Machine Learning. There are many potential societal consequences of our work, none which we feel must be specifically highlighted here.

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

# A. Proofs of Lemmas

### Proof of Lemma 4.1

*Proof.* Consider an unweighted undirected graph with adjacency $A$ and degree matrix $D = \text{diag}(d_1, \ldots, d_N)$, assuming no isolated nodes. The normalized Laplacian is:

$$L = I - D^{-1/2}AD^{-1/2}. \tag{32}$$

For any $x \in \mathbb{R}^N$, its quadratic form is:

$$x^\top L x = \sum_{u<v} A_{uv} \left( \frac{x_u}{\sqrt{d_u}} - \frac{x_v}{\sqrt{d_v}} \right)^2 \geq 0, \tag{33}$$

showing that $L$ is positive semi-definite. Taking $x = D^{1/2}\mathbf{1}$ yields $x^\top L x = 0$, so $\lambda_{\min}(L) = 0$.

To bound the largest eigenvalue, take a unit vector $x$. By $(a-b)^2 \leq 2(a^2 + b^2)$:

$$\left( \frac{x_u}{\sqrt{d_u}} - \frac{x_v}{\sqrt{d_v}} \right)^2 \leq 2 \left( \frac{x_u^2}{d_u} + \frac{x_v^2}{d_v} \right). \tag{34}$$

Summing over edges and using $A_{uv} = A_{vu}$, the two symmetric terms in the sum are equal. Extending to all pairs $(u, v)$ and using $\sum_{v:v\sim u} A_{uv} = d_u$:

$$\sum_{u<v} A_{uv} \left( \frac{x_u^2}{d_u} + \frac{x_v^2}{d_v} \right) = 2 \cdot \frac{1}{2} \sum_u x_u^2 \cdot \sum_{v:v\sim u} \frac{A_{uv}}{d_u} = \sum_u x_u^2 = \|x\|_2^2. \tag{35}$$

For any unit vector $x$, we have $x^\top L x \leq 2\|x\|_2^2 = 2$. By the Rayleigh quotient characterization of eigenvalues for symmetric matrices:

$$\lambda_{\max}(L) = \max_{x \neq 0} \frac{x^\top L x}{x^\top x} \leq 2. \tag{36}$$

Combined with the earlier lower bound, this shows that the spectrum of $L$ satisfies:

$$\Lambda(L) \subset [0, 2]. \tag{37}$$

Let $A_w = W \odot A$ with $0 \leq w_{uv} \leq 1$, and $d_w(u) = \sum_v w_{uv} A_{uv}$. The weighted normalized Laplacian is:

$$L_w = I - D_w^{-1/2} A_w D_w^{-1/2}. \tag{38}$$

Its quadratic form $x^\top L_w x = \sum_{u<v} w_{uv} A_{uv} \left( \frac{x_u}{\sqrt{d_w(u)}} - \frac{x_v}{\sqrt{d_w(v)}} \right)^2 \geq 0$, so $L_w$ is positive semi-definite. Taking $x = D_w^{1/2}\mathbf{1}$ gives $\lambda_{\min}(L_w) = 0$.

For the upper bound, since $w_{uv} \geq 0$, applying $(a-b)^2 \leq 2(a^2 + b^2)$ gives:

$$w_{uv} \left( \frac{x_u}{\sqrt{d_w(u)}} - \frac{x_v}{\sqrt{d_w(v)}} \right)^2 \leq 2w_{uv} \left( \frac{x_u^2}{d_w(u)} + \frac{x_v^2}{d_w(v)} \right). \tag{39}$$

Summing over edges, by symmetry and $\sum_{v:v\sim u} \frac{w_{uv} A_{uv}}{d_w(u)} = 1$:

$$\sum_{u<v} w_{uv} A_{uv} \left( \frac{x_u^2}{d_w(u)} + \frac{x_v^2}{d_w(v)} \right) = \sum_u x_u^2 = \|x\|_2^2. \tag{40}$$

For any unit vector $x$, we have:

$$x^\top L_w x \leq 2\|x\|_2^2 = 2. \tag{41}$$

By the Rayleigh quotient characterization of eigenvalues for symmetric matrices:

$$\lambda_{\max}(L_w) = \max_{x \neq 0} \frac{x^\top L_w x}{x^\top x} \leq 2. \tag{42}$$

Combined with the earlier observation that $\lambda_{\min}(L_w) = 0$, we conclude that the spectrum of the heterophily-weighted Laplacian satisfies:

$$\Lambda(L_w) \subset [0, 2]. \tag{43}$$

**Proof of Lemma 4.2**

*Proof.* Let $\mathcal{W} = \{W_0, W_1, \ldots, W_C\}$ denote the multi-scale Beta wavelet filter bank applied to the heterophily-weighted Laplacian $L_w$, with:

$$W_k = \beta^*_{k,C-k}(L_w), \quad \beta^*_{k,C-k}(\lambda) = \frac{(\lambda/2)^k (1 - \lambda/2)^{C-k}}{2B(k+1, C-k+1)}, \quad k = 0, 1, \ldots, C. \tag{44}$$

By Lemma 4.1, the eigenvalues of $L_w$ satisfy:

$$\Lambda(L_w) = \{\lambda_1^w, \ldots, \lambda_N^w\} \subset [0, 2]. \tag{45}$$

Since $L_w$ is symmetric and $\beta^*_{k,C-k}(\cdot)$ is a real-valued function, each $W_k$ is symmetric and diagonalizable in the eigenbasis $U$ of $L_w$:

$$W_k = U \operatorname{diag}\big(\beta^*_{k,C-k}(\lambda_1^w), \ldots, \beta^*_{k,C-k}(\lambda_N^w)\big)U^\top, \quad U^\top U = I. \tag{46}$$

Define:

$$G(\lambda) = \sum_{k=0}^{C} \big(\beta^*_{k,C-k}(\lambda)\big)^2. $$

Consider the frame operator:

$$S = \sum_{k=0}^{C} W_k^\top W_k = \sum_{k=0}^{C} W_k^2 = U \operatorname{diag}\big(G(\lambda_1^w), \ldots, G(\lambda_N^w)\big)U^\top. \tag{47}$$

The multi-scale Beta wavelet filter bank consists of endpoint filters ($k = 0$ and $k = C$) and intermediate filters ($0 < k < C$). For any $\lambda \in (0, 2)$, all responses $\beta^*_{k,C-k}(\lambda)$ are strictly positive. At the endpoints, the endpoint filters satisfy $\beta^*_{0,C}(0) = 1/(2B(1, C+1)) > 0$ and $\beta^*_{C,0}(2) = 1/(2B(C+1, 1)) > 0$. Hence, for every eigenvalue $\lambda_\ell^w \in [0, 2]$, there exists at least one $k$ such that:

$$\beta^*_{k,C-k}(\lambda_\ell^w) > 0, \tag{48}$$

ensuring that all diagonal entries of $S$ in the eigenbasis are strictly positive.

Define:

$$A = \min_{\lambda \in [0,2]} \sum_{k=0}^{C} (\beta^*_{k,C-k}(\lambda))^2, \quad B = \max_{\lambda \in [0,2]} \sum_{k=0}^{C} (\beta^*_{k,C-k}(\lambda))^2. \tag{49}$$

Since $G(\lambda)$ is continuous on the compact interval $[0, 2]$ and satisfies $G(\lambda) > 0$ for all $\lambda \in [0, 2]$, we have:

$$0 < A \leq B < \infty. \tag{50}$$

For any $x \in \mathbb{R}^N$:

$$\sum_{k=0}^{C} \|W_k x\|_2^2 = \sum_{k=0}^{C} x^\top W_k^\top W_k x = x^\top S x. \tag{51}$$

Since the eigenvalues of $S$ are given by $G(\lambda_1^w), \ldots, G(\lambda_N^w)$ and satisfy $A \leq G(\lambda_\ell^w) \leq B$, we obtain:

$$A\|x\|_2^2 \leq \sum_{k=0}^{C} \|W_k x\|_2^2 = x^\top S x \leq B\|x\|_2^2. \tag{52}$$

Therefore, $\mathcal{W}$ forms a stable graph wavelet frame for the heterophily-weighted graph.

**Proof of Lemma 4.3**

*Proof.* We expand the $\kappa$-addition for small vectors $\mathbf{x}, \mathbf{y} \in \mathbb{R}^h$, keeping terms up to first order in $\kappa$ and retaining the leading $\kappa$-linear cubic terms in $\mathbf{x}, \mathbf{y}$.

The remainder term $R_\kappa(\mathbf{x}, \mathbf{y})$ collects higher-order terms. More precisely, the calculation below yields:

$$R_\kappa(\mathbf{x}, \mathbf{y}) = O(\kappa^2 \rho^5), \qquad \rho = \|\mathbf{x}\| + \|\mathbf{y}\|.$$

By definition:

$$\mathbf{x} \oplus_\kappa \mathbf{y} = \frac{(1 - 2\kappa\langle\mathbf{x}, \mathbf{y}\rangle - \kappa\|\mathbf{y}\|^2)\mathbf{x} + (1 + \kappa\|\mathbf{x}\|^2)\mathbf{y}}{1 - 2\kappa\langle\mathbf{x}, \mathbf{y}\rangle + \kappa^2\|\mathbf{x}\|^2\|\mathbf{y}\|^2}.$$

Expanding the denominator gives:

$$\frac{1}{1 - 2\kappa\langle\mathbf{x}, \mathbf{y}\rangle + \kappa^2\|\mathbf{x}\|^2\|\mathbf{y}\|^2} = 1 + 2\kappa\langle\mathbf{x}, \mathbf{y}\rangle + O(\kappa^2 \rho^4).$$

Therefore:

$$\mathbf{x} \oplus_\kappa \mathbf{y} = \left[(1 - 2\kappa\langle\mathbf{x}, \mathbf{y}\rangle - \kappa\|\mathbf{y}\|^2)\mathbf{x} + (1 + \kappa\|\mathbf{x}\|^2)\mathbf{y}\right]\left[1 + 2\kappa\langle\mathbf{x}, \mathbf{y}\rangle\right] + O(\kappa^2 \rho^5)$$

$$= \mathbf{x} + \mathbf{y} + \kappa\left(\|\mathbf{x}\|^2\mathbf{y} - \|\mathbf{y}\|^2\mathbf{x} + 2\langle\mathbf{x}, \mathbf{y}\rangle\mathbf{y}\right) + O(\kappa^2 \rho^5).$$

Thus the stated expansion follows by setting:

$$R_\kappa(\mathbf{x}, \mathbf{y}) = O(\kappa^2 \rho^5).$$

The displayed $\kappa$-linear correction is not symmetric in $\mathbf{x}$ and $\mathbf{y}$. Hence the operation is generically non-commutative for $\kappa \neq 0$.

## B. Dataset Statistics

In this appendix, we summarize the basic statistics of the six real-world datasets used in our experiments for GAD in Table 3.

*Table 3.* Dataset statistics for the experiments. Domain indicates the application area, $|\mathcal{V}|$ and $|\mathcal{E}|$ denote the numbers of nodes and edges, respectively; $d$ denotes the node attribute dimension. Anomaly % denotes the proportion of anomalous nodes, Edge Heterophily % measures the proportion of edges connecting nodes with different labels, and Average Degree denotes the average number of edges per node, computed as $|\mathcal{E}|/|\mathcal{V}|$.

| Dataset | Domain | $|\mathcal{V}|$ | $|\mathcal{E}|$ | $d$ | Anomaly % | Edge Heterophily % | Average Degree |
|---|---|---|---|---|---|---|---|
| Weibo | Under Same Hashtag | 8,405 | 407,963 | 400 | 10.33% | 2.34% | 48.54 |
| Tolokers | Work Collaboration | 11,758 | 519,000 | 10 | 21.82% | 40.55% | 44.14 |
| Amazon | Review Correlation | 11,944 | 4,398,392 | 25 | 6.87% | 4.62% | 368.25 |
| Yelp | Reviewer Interaction | 45,954 | 3,846,979 | 32 | 14.53% | 22.70% | 83.71 |
| T-Finance | Transaction Record | 39,357 | 21,222,543 | 10 | 4.58% | 2.92% | 539.23 |
| T-Social | Social Friendship | 5,781,065 | 73,105,508 | 10 | 3.01% | 37.61% | 12.64 |

We also compute the node-level heterophily ratios for both normal and anomalous nodes across six real-world datasets, and report the corresponding density distributions in Figure 5. The distributions reveal a systematic rightward shift for anomalous nodes, indicating higher proportions of heterophilous edges in their local neighborhoods.

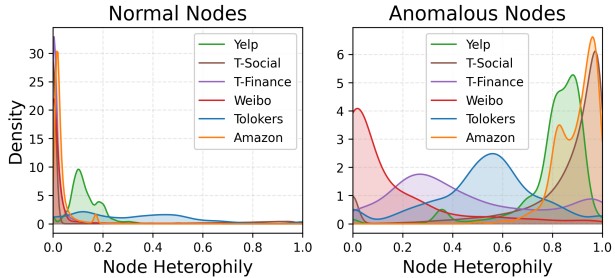

*Figure 5.* Density distributions of node-level heterophily ratios for normal and anomalous nodes across six datasets. Anomalous nodes exhibit higher heterophily ratios, with distributions shifted to the right.

## C. Evaluation Metrics

In this appendix, we describe the evaluation metrics used to assess anomaly detection performance, including AUROC and AUPRC, which are threshold-independent, and F1-Macro and G-Mean, which require threshold selection and capture precision-recall trade-offs as well as class-wise balance.

**AUROC.** The Receiver Operating Characteristic (ROC) curve is defined by the true positive rate (TPR) and false positive rate (FPR) as functions of the decision threshold $\tau$:

$$\text{TPR}(\tau) = \frac{\sum_i \mathbb{I}(s_i \geq \tau,\, y_i = 1)}{\sum_i \mathbb{I}(y_i = 1)}, \quad \text{FPR}(\tau) = \frac{\sum_i \mathbb{I}(s_i \geq \tau,\, y_i = 0)}{\sum_i \mathbb{I}(y_i = 0)}. \tag{53}$$

AUROC is defined as the area under the ROC curve:

$$\text{AUROC} = \int_0^1 \text{TPR}(u)\, d\,\text{FPR}(u), \tag{54}$$

which is equivalent to the probability that a randomly chosen positive sample is assigned a higher score than a randomly chosen negative sample.

**AUPRC.** The Precision–Recall (PR) curve is defined by precision and recall as functions of $\tau$:

$$\text{Precision}(\tau) = \frac{\sum_i \mathbb{I}(s_i \geq \tau,\, y_i = 1)}{\sum_i \mathbb{I}(s_i \geq \tau)}, \quad \text{Recall}(\tau) = \frac{\sum_i \mathbb{I}(s_i \geq \tau,\, y_i = 1)}{\sum_i \mathbb{I}(y_i = 1)}. \tag{55}$$

AUPRC is defined as the area under the PR curve:

$$\text{AUPRC} = \int_0^1 \text{Precision}(r)\, d\,\text{Recall}(r). \tag{56}$$

**F1-Macro.** Given a threshold $\tau$, let $\hat{y}_i = \mathbb{I}(s_i \geq \tau)$. For each class $c \in \{0, 1\}$, the precision and recall are defined as:

$$\text{Precision}_c(\tau) = \frac{\sum_i \mathbb{I}(\hat{y}_i = c,\, y_i = c)}{\sum_i \mathbb{I}(\hat{y}_i = c)}, \quad \text{Recall}_c(\tau) = \frac{\sum_i \mathbb{I}(\hat{y}_i = c,\, y_i = c)}{\sum_i \mathbb{I}(y_i = c)}. \tag{57}$$

The F1 score for class $c$ is:

$$\text{F1}_c(\tau) = 2 \cdot \frac{\text{Precision}_c(\tau) \cdot \text{Recall}_c(\tau)}{\text{Precision}_c(\tau) + \text{Recall}_c(\tau)}. \tag{58}$$

F1-Macro is the unweighted average over both classes:

$$\text{F1-Macro}(\tau) = \frac{1}{2}\left(\text{F1}_0(\tau) + \text{F1}_1(\tau)\right). \tag{59}$$

**G-Mean.** Given a threshold $\tau$, let $\hat{y}_i = \mathbb{I}(s_i \geq \tau)$. The recall for class 0 and class 1 is:

$$\text{Recall}_0(\tau) = \frac{\sum_i \mathbb{I}(\hat{y}_i = 0, \, y_i = 0)}{\sum_i \mathbb{I}(y_i = 0)}, \quad \text{Recall}_1(\tau) = \frac{\sum_i \mathbb{I}(\hat{y}_i = 1, \, y_i = 1)}{\sum_i \mathbb{I}(y_i = 1)}. \tag{60}$$

G-Mean is defined as the geometric mean of the recalls:

$$\text{G-Mean}(\tau) = \sqrt{\text{Recall}_0(\tau) \cdot \text{Recall}_1(\tau)}. \tag{61}$$

**Threshold Selection.** In our experiments, the decision threshold $\tau^*$ is selected based on the validation set:

$$\tau^* = \arg\max_{\tau} \text{F1-Macro}_{\text{val}}(\tau). \tag{62}$$

This threshold $\tau^*$ is then applied to the test set predictions to compute F1-Macro and G-Mean.

## D. Algorithm

---

**Algorithm 1** Heterophily-Weighted Spectral Filtering (HWSF)

---

1: **Input:** Graph $G = (V, E, \mathbf{X})$, edge predictor $g_\phi$, learnable scaling parameter $\alpha_w$, heterophily quantile threshold $\tau$, Beta filter order $C$, fusion coefficient $\rho$
2: **Output:** Spectral node representations $\mathbf{H}^s$
3: **for** each edge $(i, j) \in E$ **do**
4:      Edge feature: $\mathbf{x}_{ij} \leftarrow |\mathbf{x}_i - \mathbf{x}_j|$
5:      Predict heterophily score: $\hat{e}_{ij} \leftarrow \sigma(g_\phi(\mathbf{x}_{ij}))$
6:      Compute edge weight: $w_{ij} \leftarrow \exp(-\alpha_w \hat{e}_{ij})$
7: **end for**
8: Construct weighted adjacency matrix $A_w$
9: Construct heterophily masks: $\mathcal{M}^{\text{low}} \leftarrow \mathbb{I}(\hat{e}_{ij} \leq \tau), \quad \mathcal{M}^{\text{high}} \leftarrow \mathbb{I}(\hat{e}_{ij} > \tau)$
10: Apply masks to adjacency: $A_w^{\text{low}} \leftarrow \mathcal{M}^{\text{low}} \odot A_w, \quad A_w^{\text{high}} \leftarrow \mathcal{M}^{\text{high}} \odot A_w$
11: Compute Laplacians $L_w^{\text{low}}, L_w^{\text{high}}$
12: Project node attributes: $\mathbf{H}_0 \leftarrow \text{MLP}(\mathbf{X})$
13: **for** $m \in \{\text{low}, \text{high}\}$ **do**
14:      **for** $k = 0$ to $C$ **do**
15:          Construct Beta filter: $W_k^m \leftarrow \beta_{k,C-k}^*(L_w^m)$
16:          Filter features: $\mathbf{H}_k^m \leftarrow W_k^m \mathbf{H}_0$
17:      **end for**
18:      Concatenate multi-scale features: $\mathbf{H}^m \leftarrow [\mathbf{H}_0^m, \mathbf{H}_1^m, \ldots, \mathbf{H}_C^m]$
19: **end for**
20: Fuse $\mathbf{H}^{\text{low}}$ and $\mathbf{H}^{\text{high}}$ using the gate coefficient $\rho$
21: Apply a linear projection to the fused representation to obtain $\mathbf{H}^s$
22: **Return:** $\mathbf{H}^s$

---

In this appendix, we provide the pseudocode for two key modules of HSMAD. These procedures are summarized in Algorithms 1 and 2, respectively.

## E. Supplementary Performance Analysis

In this appendix, we provide supplementary performance comparisons across six datasets using AUPRC and G-Mean as additional evaluation metrics, as reported in Table 4.

## F. Supplementary Ablation Study

In this appendix, we provide supplementary ablation results for the HSMAD framework using AUPRC and G-Mean as additional evaluation metrics across six datasets, as reported in Table 5.

---

**Algorithm 2** Heterophily-Routed Manifold Update (HRMU)

---

1: **Input:** Graph $G = (V, E, \mathbf{X})$, learnable curvatures $\kappa^+ > 0$, $\kappa^- < 0$, heterophily scores $\hat{e}_{ji}$, learnable scaling parameter $\alpha$, fusion coefficient $\gamma$
2: **Output:** Manifold node representations $\mathbf{H^c}$
3: Initialize: $\mathbf{H}^{(0)} \leftarrow \mathbf{X}$
4: **for** layer $l = 0$ to $L - 1$ **do**
5:    **for** each node $i \in \{1, \ldots, N\}$ **do**
6:       Transform self-feature: $\tilde{\mathbf{h}}_i^{(l)} \leftarrow (W_0^{(l)})^\top \mathbf{h}_i^{(l)}$
7:       **for** each neighbor $j \in \mathcal{N}(i)$ **do**
8:          Transform neighbor feature: $\tilde{\mathbf{h}}_j^{(l)} \leftarrow \frac{1}{\sqrt{d_j}}(W^{(l)})^\top \mathbf{h}_j^{(l)}$
9:          Compute heterophily routing weights: $w_{ji}^+ \leftarrow \exp(-\alpha \hat{e}_{ji})$,    $w_{ji}^- \leftarrow 1 - \exp(-\alpha \hat{e}_{ji})$
10:      **end for**
11:      Aggregate messages for each curvature channel: $\mathbf{m}_i^{(l,+)} \leftarrow \sum_{j \in \mathcal{N}(i)} w_{ji}^+ \tilde{\mathbf{h}}_j^{(l)}$,    $\mathbf{m}_i^{(l,-)} \leftarrow \sum_{j \in \mathcal{N}(i)} w_{ji}^- \tilde{\mathbf{h}}_j^{(l)}$
12:      Positive-curvature update ($\kappa^+ > 0$): $\mathbf{h}_i^{(l+1,+)} \leftarrow \log_{\kappa^+}^{\mathbf{0}} \left( \exp_{\kappa^+}^{\mathbf{0}}(\tilde{\mathbf{h}}_i^{(l)}) \oplus_{\kappa^+} \exp_{\kappa^+}^{\mathbf{0}}(\mathbf{m}_i^{(l,+)}) \right)$
13:      Negative-curvature update ($\kappa^- < 0$): $\mathbf{h}_i^{(l+1,-)} \leftarrow \log_{\kappa^-}^{\mathbf{0}} \left( \exp_{\kappa^-}^{\mathbf{0}}(\tilde{\mathbf{h}}_i^{(l)}) \oplus_{\kappa^-} \exp_{\kappa^-}^{\mathbf{0}}(\mathbf{m}_i^{(l,-)}) \right)$
14:    **end for**
15:    Update channel-wise representations: $\mathbf{H}^{(l+1,+)} \leftarrow \{\mathbf{h}_i^{(l+1,+)}\}_{i=1}^N$,    $\mathbf{H}^{(l+1,-)} \leftarrow \{\mathbf{h}_i^{(l+1,-)}\}_{i=1}^N$
16: **end for**
17: **for** each node $i \in \{1, \ldots, N\}$ **do**
18:    Fuse two channels: $\mathbf{h}_i \leftarrow \gamma \mathbf{h}_i^{(L,+)} + (1 - \gamma) \mathbf{h}_i^{(L,-)}$
19: **end for**
20: **Return:** $\mathbf{H^c}$

---

*Table 4.* Performance comparison of all datasets in terms of AUPRC and G-Mean. Bold and underlined values indicate the best and second-best results, respectively. "OOM" denotes out of memory, and "OOT" denotes out of time (running longer than one day).

| Method | Amazon | | Yelp | | Weibo | | Tolokers | | T-Finance | | T-Social | | Avg. | |
|---|---|---|---|---|---|---|---|---|---|---|---|---|---|---|
| | AUPRC | G-Mean | AUPRC | G-Mean | AUPRC | G-Mean | AUPRC | G-Mean | AUPRC | G-Mean | AUPRC | G-Mean | AUPRC | G-Mean |
| MLP | 0.8887 | 0.8964 | 0.4775 | 0.6670 | 0.8631 | 0.8708 | 0.3923 | 0.6125 | 0.6885 | 0.7702 | 0.0686 | 0.4987 | 0.5631 | 0.7193 |
| GCN | 0.2902 | 0.6481 | 0.2378 | 0.4415 | 0.9472 | 0.9473 | 0.4426 | 0.6214 | 0.7177 | 0.8020 | 0.3187 | 0.5295 | 0.4924 | 0.6650 |
| GAT | 0.8819 | 0.8972 | 0.4422 | 0.6592 | 0.9143 | 0.9351 | 0.4457 | 0.6781 | 0.6327 | 0.8188 | 0.1300 | 0.4757 | 0.5745 | 0.7440 |
| GraphSAGE | 0.7007 | 0.8141 | 0.5400 | 0.7012 | 0.8213 | 0.9053 | 0.4624 | 0.6766 | 0.2587 | 0.6656 | 0.0778 | 0.3873 | 0.4768 | 0.6917 |
| AMNet | 0.8577 | 0.8696 | 0.5199 | 0.5637 | 0.8936 | 0.8701 | 0.3443 | 0.0684 | 0.7444 | 0.7295 | 0.0202 | 0.0814 | 0.5634 | 0.5305 |
| BWGNN | 0.8834 | 0.9075 | 0.5852 | 0.7072 | 0.9205 | 0.9326 | 0.4986 | 0.6627 | 0.8630 | 0.8694 | 0.8244 | 0.8594 | 0.7625 | 0.8231 |
| GHRN | 0.8897 | 0.9044 | 0.5770 | 0.6984 | 0.9021 | 0.8998 | 0.4687 | 0.6816 | 0.7864 | 0.8027 | 0.8767 | 0.8772 | 0.7501 | 0.8107 |
| SparseGAD | 0.8591 | 0.8678 | 0.5693 | 0.6053 | 0.8864 | 0.9231 | 0.4296 | 0.3022 | 0.8045 | 0.8430 | OOM | OOM | 0.7098 | 0.7083 |
| SEC-GFD | 0.8840 | 0.8941 | 0.5896 | 0.7238 | 0.9112 | 0.9279 | 0.4903 | 0.6738 | 0.8374 | 0.8648 | 0.7711 | 0.8397 | 0.7473 | 0.8207 |
| NRGL | 0.8479 | 0.9013 | 0.3883 | 0.5944 | 0.7216 | 0.8285 | 0.3840 | 0.6216 | 0.5528 | 0.8068 | OOM | OOM | 0.5789 | 0.7505 |
| PC-GNN | 0.8437 | 0.8797 | 0.3690 | 0.6686 | 0.7953 | 0.8257 | 0.3774 | 0.6160 | 0.6821 | 0.8164 | 0.0825 | 0.6087 | 0.5250 | 0.7359 |
| ConsisGAD | 0.8684 | 0.8842 | 0.5098 | 0.6850 | 0.9019 | 0.8999 | 0.4178 | 0.6722 | 0.8851 | 0.8792 | 0.6894 | 0.8025 | 0.7121 | 0.8038 |
| PMP | 0.9057 | 0.8622 | 0.7744 | 0.8233 | 0.9369 | 0.9321 | 0.5182 | 0.6485 | 0.8874 | 0.8862 | 0.9452 | 0.9208 | 0.8280 | 0.8455 |
| DSGAD | 0.8939 | 0.8934 | 0.5645 | 0.6947 | 0.9318 | 0.9369 | 0.4951 | 0.6793 | 0.8797 | 0.8873 | 0.8719 | 0.8841 | 0.7728 | 0.8293 |
| CurvGAD | 0.8607 | 0.8893 | 0.3652 | 0.5887 | 0.8042 | 0.8224 | 0.3907 | 0.6391 | 0.8468 | 0.8702 | OOM | OOM | 0.6535 | 0.7619 |
| SpaceGNN | 0.9006 | 0.9026 | 0.6879 | 0.7514 | 0.9409 | 0.9375 | 0.4788 | 0.5954 | 0.8984 | 0.8929 | OOT | OOT | 0.7813 | 0.8160 |
| CGADM | 0.8853 | 0.8877 | 0.8046 | 0.8292 | 0.9577 | 0.9499 | 0.4617 | 0.6553 | 0.7878 | 0.8504 | **0.9517** | 0.9355 | 0.8081 | 0.8513 |
| HSMAD | **0.9213** | **0.9250** | **0.8377** | **0.8435** | **0.9679** | **0.9518** | **0.5859** | **0.7155** | **0.9037** | **0.8988** | 0.9489 | **0.9423** | **0.8609** | **0.8795** |

*Table 5.* Ablation study on six datasets in terms of AUPRC and G-Mean. "w/o" denotes removing the corresponding component.

| Method | Amazon | | Yelp | | Weibo | | Tolokers | | T-Finance | | T-Social | | Avg. | |
|---|---|---|---|---|---|---|---|---|---|---|---|---|---|---|
| | AUPRC | G-Mean | AUPRC | G-Mean | AUPRC | G-Mean | AUPRC | G-Mean | AUPRC | G-Mean | AUPRC | G-Mean | AUPRC | G-Mean |
| w/o HWSF | 0.9070 | 0.9225 | 0.7779 | 0.8072 | 0.9637 | 0.9507 | 0.5703 | 0.7114 | 0.8873 | 0.8807 | 0.9052 | 0.9096 | 0.8352 | 0.8637 |
| w/o HRMU | 0.9153 | 0.9246 | 0.7597 | 0.8289 | 0.9562 | 0.9507 | 0.5743 | 0.6869 | 0.8863 | 0.8892 | 0.8840 | 0.8931 | 0.8293 | 0.8622 |
| HSMAD | **0.9213** | **0.9250** | **0.8377** | **0.8435** | **0.9679** | **0.9518** | **0.5859** | **0.7155** | **0.9037** | **0.8988** | **0.9489** | **0.9423** | **0.8609** | **0.8795** |

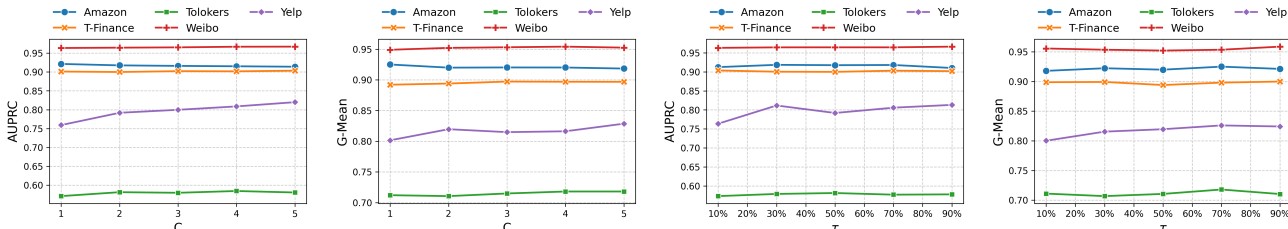

*Figure 6.* Sensitivity analysis of HSMAD with respect to the spectral filter order $C$ and the quantile $\tau$. Top row: AUPRC and G-Mean on Amazon, Yelp, Weibo, Tolokers, and T-Finance with fixed $\tau = 0.5$ and varying $C$. Bottom row: AUPRC and G-Mean on the same datasets with fixed $C = 2$ and varying $\tau$.

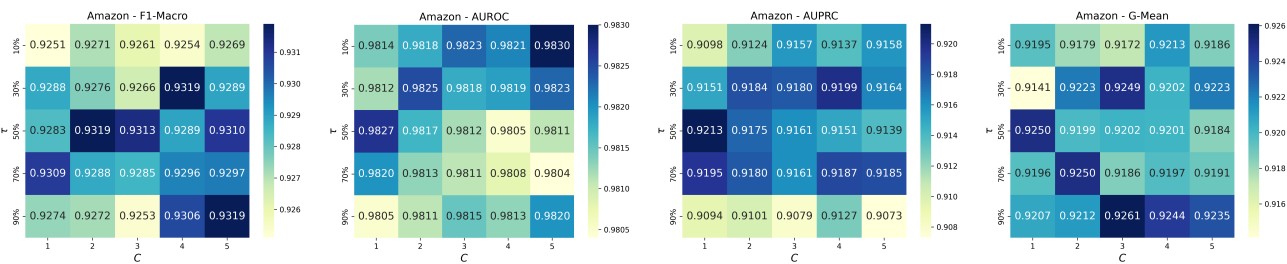

*Figure 7.* Sensitivity analysis of HSMAD on Amazon dataset. The horizontal axis corresponds to the spectral filter order $C$ (ranging from 1 to 5), and the vertical axis corresponds to the quantile threshold $\tau$ (ranging from 10% to 90% with steps of 20%).

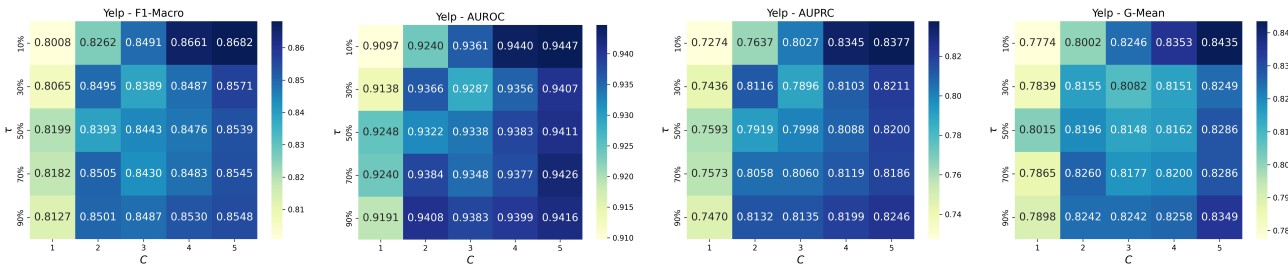

*Figure 8.* Sensitivity analysis of HSMAD on Yelp dataset. The horizontal axis corresponds to the spectral filter order $C$ (ranging from 1 to 5), and the vertical axis corresponds to the quantile threshold $\tau$ (ranging from 10% to 90% with steps of 20%).

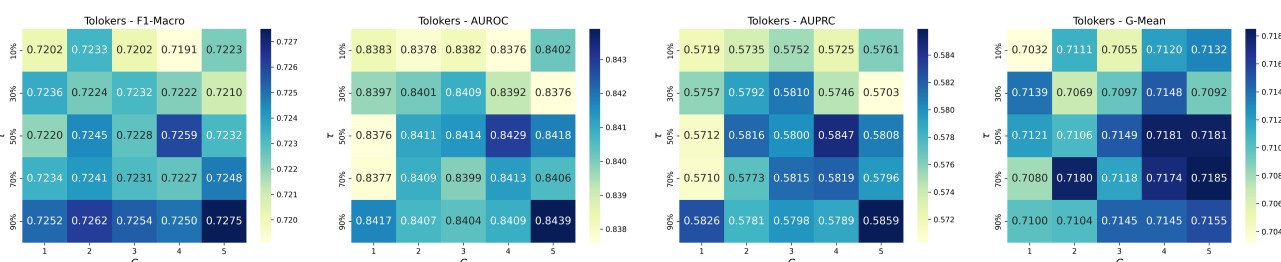

*Figure 9.* Sensitivity analysis of HSMAD on Tolokers dataset. The horizontal axis corresponds to the spectral filter order $C$ (ranging from 1 to 5), and the vertical axis corresponds to the quantile threshold $\tau$ (ranging from 10% to 90% with steps of 20%).

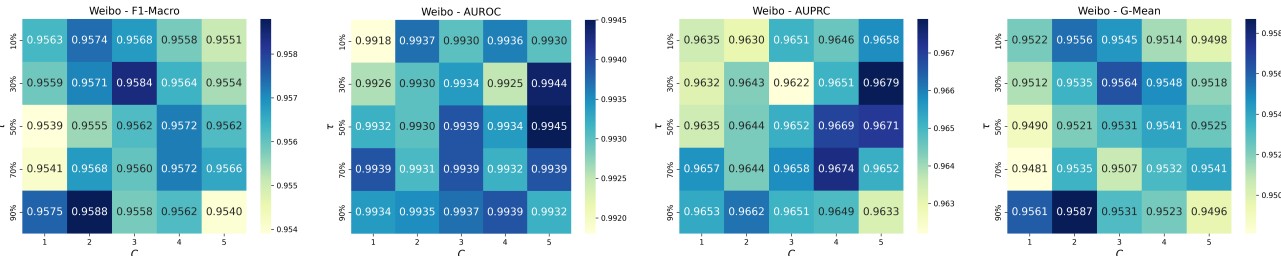

*Figure 10.* Sensitivity analysis of HSMAD on Weibo dataset. The horizontal axis corresponds to the spectral filter order $C$ (ranging from 1 to 5), and the vertical axis corresponds to the quantile threshold $\tau$ (ranging from 10% to 90% with steps of 20%).

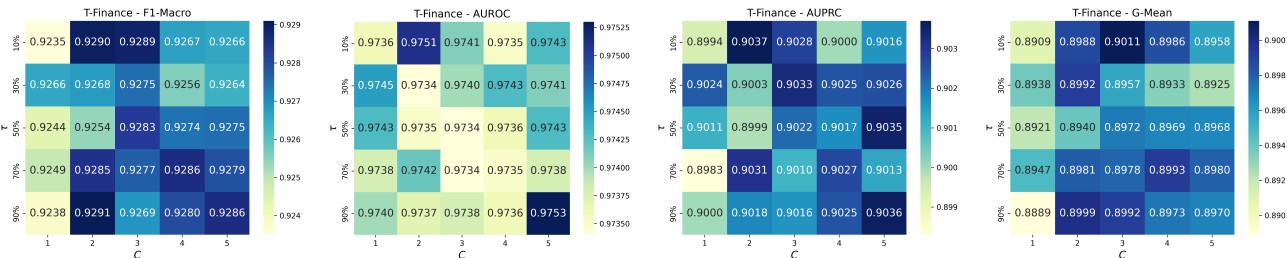

*Figure 11.* Sensitivity analysis of HSMAD on T-Finance dataset. The horizontal axis corresponds to the spectral filter order $C$ (ranging from 1 to 5), and the vertical axis corresponds to the quantile threshold $\tau$ (ranging from 10% to 90% with steps of 20%).

## G. Supplementary Sensitivity Analysis

In this appendix, we provide a sensitivity analysis of HSMAD with respect to the spectral filter order $C$ and the quantile threshold $\tau$ across five real-world datasets. The AUPRC and G-Mean results are shown in Figure 6, and the dataset-specific heatmaps are presented in Figures 7, 8, 9, 10, and 11.

## H. Supplementary Complexity Analysis

*Table 6.* Efficiency comparison on Yelp and T-Social. Training time is measured in seconds per epoch, inference time is measured in seconds per evaluation, and memory usage is reported in MB. "OOM" denotes out of memory, and "OOT" denotes out of time (running longer than one day).

| Method | Yelp | | | | T-Social | | | |
|---|---|---|---|---|---|---|---|---|
| | Training Time | Inference Time | Training Memory | Inference Memory | Training Time | Inference Time | Training Memory | Inference Memory |
| DSGAD | 0.0909 | 0.0060 | 1193.47 | 913.70 | 2.4341 | 0.0157 | 23268.58 | 17741.09 |
| CurvGAD | 4.8877 | 4.0130 | 11142.37 | 3361.36 | - | - | OOM | OOM |
| SpaceGNN | 11.0786 | 2.2711 | 459.00 | 334.94 | OOT | OOT | - | - |
| CGADM | 2.4930 | 2.1519 | 4788.29 | 4211.10 | 13.9265 | 15.4230 | 32495.69 | 20639.04 |
| HSMAD | 0.4720 | 0.2802 | 5397.17 | 3386.15 | 9.9169 | 6.4775 | 65100.89 | 23613.84 |

In this appendix, we provide a supplementary analysis of the time and space complexity of HSMAD. The framework mainly consists of two components: Heterophily-Weighted Spectral Filtering and Heterophily-Routed Manifold Update. HWSF constructs edge-level representations and predicts edge heterophily through an MLP. Compared with standard GNN layers that mainly store node features and graph connectivity, HWSF additionally maintains edge features and filtering-related intermediate variables. Therefore, its space cost increases with the number of edges. The multi-band spectral filtering operation also introduces additional computation, since multiple spectral bands are used to capture graph signals at different frequency ranges. HRMU performs weighted message passing followed by manifold-based feature updates. Compared with conventional Euclidean message passing, the $\kappa$-Addition operation requires additional computations associated with manifold geometry. As a result, HRMU introduces extra computational overhead, especially when applied to large graphs with high-dimensional node attributes.

To further assess practical efficiency, we conduct empirical comparisons with recent methods, including DSGAD, CurvGAD, SpaceGNN, and CGADM. The results on Yelp and T-Social are reported in Table 6. The comparison includes training time, inference time, training memory, and inference memory. The additional overhead in HSMAD arises from edge-wise feature modeling and manifold geometry operations, which are specifically designed to capture heterophily and achieve strong performance. These modules inevitably increase computational and memory requirements, but HSMAD remains more resource-efficient than methods such as CurvGAD that also employ manifold representations. We acknowledge this trade-off and plan to further explore resource-efficient strategies in future work to reduce computational and memory costs while retaining performance.

