# OpenReview forum: "HSMAD: Heterophily-Driven Spectral and Manifold Learning for Graph Anomaly Detection"
_ICML.cc/2026/Conference — ICML 2026 regular_

### Official Review · Reviewer_6419 · 2026-03-02

**Soundness:** 3
**Presentation:** 3
**Significance:** 3
**Originality:** 3
**Overall Recommendation:** 4
**Confidence:** 4

**Summary:**

HSMAD deals with node-level graph anomaly detection under heterophily, where neighbors often have dissimilar labels/semantics and thus standard homophily-biased GNN message passing and common spectral filters can over-smooth or dilute anomaly signals. The method combines two complementary components: HWSF (Heterophily-Weighted Spectral Filtering), which predicts edge-level heterophily scores and uses them to construct a weighted Laplacian and a heterophily mask, then applies a multi-band Beta-wavelet filter bank to extract frequency-aware representations that preserve heterophily-induced discriminative patterns; and HRMU (Heterophily-Routed Manifold Update), which routes messages according to heterophily strength into different curvature manifolds (positive vs. negative curvature) and performs manifold-aware updates before fusing curvature-specific embeddings. The final anomaly scoring model integrates spectral and manifold embeddings and is trained with a joint objective that includes a node classification loss and a heterophily prediction loss. Experiments on six real-world datasets report consistent improvements in F1-Macro and AUROC over a broad set of anomaly detection baselines, with ablations indicating both modules contribute. The paper’s main technical claim is that explicitly modeling heterophily in both spectral and geometric domains yields more robust anomaly representations than relying on implicit homophily assumptions.

**Compliance With Llm Reviewing Policy:**

Affirmed.

**Key Questions For Authors:**

1.How does the edge predictor handle edges between two unlabeled nodes during training? Does the limited availability of node labels impact the accuracy of spectral re-weighting? \
2.In the HRMU module, how are the initial curvatures \kappa^+ and \kappa^- determined, and how sensitive is the model to these initializations? \
3.Can the authors provide a memory/time complexity comparison specifically for the manifold addition operation versus standard GCN message passing on the T-Social dataset?

**Limitations:**

NO. Although, the authors acknowledge that HSMAD may struggle with extremely sparse graphs where neighborhood signals are insufficient for accurate heterophily prediction. The computational overhead of manifold operations is noted as a potential bottleneck for extremely large graphs. Additionally, the societal impact discussion is somewhat generic and could be expanded to address potential biases in fraud detection.

**Strengths And Weaknesses:**

Strengths:\
1.HSMAD combines spectral filtering and manifold learning in one framework. This dual-domain approach handles graph heterophily well. It is a new way to detect anomalies in complex graphs.\
2.The authors provide solid math proofs. Lemma 4.1 to 4.3 show the stability of the model. The theory supports the framework design very well.\
3.Experiments prove the model can scale to the large T-Social dataset. This dataset has over five million nodes. Most existing graph anomaly detection models fail on such large graphs.\
4.The heterophily-weighted Laplacian helps concentrate spectral energy into specific eigenmodes. Figure 4 shows this effect clearly. This allows Beta wavelet filters to find anomaly signals much more easily.\
5.HSMAD achieves state-of-the-art results on six real-world datasets. The average F1-Macro score improves by 2.66 percent. This shows the framework is very effective for practical fraud detection tasks.\

Weaknesses:\
1.The edge predictor training requires ground-truth heterophily labels. Real-world graphs often have very few labels. The authors should explain how the model works when most node labels are missing.\
2.The spectral module uses Beta wavelets similar to the BWGNN model. Using edge weights is a small change. This makes the spectral part of the contribution feel somewhat incremental.\
3.Equation 23 does not explain why the tangent space origin is the best choice. The geometric intuition for this specific projection is missing. More details would help readers understand.\

Overall, the submission is technically sound, the presentation is very clear, the impact on large-scale GAD is significant, and the dual-domain integration shows good originality.

---

> ### Author Rebuttal · Authors · 2026-03-30
>
> We sincerely thank the reviewer's valuable suggestions and careful examination of our work.
>
> **W1 and Q1. Handling limited node labels**
>
> Our method operates in a supervised setting and relies on node labels to generate edge heterophily supervision. Consistent with other label-dependent supervised baselines, the model performs best when a majority of nodes are labeled, as sufficient node labels provide the necessary supervision for accurate edge prediction. Edge heterophily labels are determined based on the availability of node labels. For edges where both nodes lack labels, the heterophily label cannot be assigned. During training, the model still computes heterophily scores for these edges, but they are excluded from the training loss. These edges remain available for heterophily prediction at inference time.
>
> As a supervised method, HSMAD shares a common limitation with other label-dependent models: in scenarios of extreme label sparsity or fully unlabeled graphs, there is insufficient supervision to guide learning, which may reduce the accuracy of the edge predictor and the resulting spectral re-weighting. To address this limitation, we plan to explore future directions such as contrastive learning with dual-domain representations and other self-supervised strategies, aiming to enhance robustness under conditions of extreme label scarcity.
>
> **W2. Incremental contribution of the spectral module**
>
> Our spectral module is inspired by BWGNN’s multi-band wavelet filtering. While the wavelet mechanism is similar, our main contribution is the heterophily-aware edge weighting, which emphasizes structural patterns relevant for anomaly detection and captures heterophily-specific signals that standard wavelet filtering may miss.
>
> We integrate this spectral module with a heterophily-routed manifold update module, forming a dual-domain architecture. This combination enhances the robustness of node representations under heterophilic graphs, supporting more accurate anomaly detection.
>
> We appreciate the reviewer's perspective and acknowledge that exploring novel spectral designs specifically for heterophilic anomaly detection is a promising direction for future work.
>
> **W3. Equation 23 does not explain the choice of tangent space origin**
>
> We choose the origin of the tangent space as the base point because it simplifies the exponential and logarithmic maps to radial scaling operations, improving geometric interpretability and numerical stability. Using an arbitrary point could introduce position-dependent distortion, varying conformal factors, and potential instability near the manifold boundary.
>
> The origin also provides a unified global tangent space for all nodes, enabling consistent message aggregation without requiring parallel transport between local tangent spaces. This design aligns with standard practice in Constant curvature graph convolutional networks (ICML 2020) and ensures that our model naturally recovers Euclidean GNNs when curvature vanishes.
>
> We will add this explanation in the revised manuscript to clarify the choice of the tangent space origin.
>
> **Q2. Initialization of curvatures**
>
> In the HRMU module, the curvature parameters $\kappa^+$ and $\kappa^-$ are learnable. Both are initialized near zero and passed through a scaled $\tanh$ function to ensure they are positive. The parameters are then assigned positive and negative signs to represent $\kappa^+$ and $\kappa^-$. This initialization provides a neutral starting point for training, allowing the model to learn curvature values appropriate for each dataset.
>
> We will add this explanation in the revised manuscript to clarify the initialization of these parameters.
>
> **Q3. Provide a memory/time complexity comparison specifically for the manifold addition operation versus standard GCN message passing on the T-Social dataset**
>
> To assess the computational overhead of manifold-based addition, we compare a standard two-layer GCN with the same GCN augmented with manifold addition on the T-Social dataset:
>
> |Method|Training Time (s/epoch)|Inference Time (s/eval)|Train Memory (MB)|Inference Memory (MB)|
> |------|------------------------|------------------------|----------------------|--------------------------|
> |GCN|0.4997|0.2465|3412.16|2949.04|
> |Manifold|0.6642|0.4427|10193.55|3505.89|
>
> Incorporating manifold addition introduces additional time and memory overhead due to the exponential and logarithmic map operations required for message passing. We acknowledge this cost, and plan to explore more efficient approximations of manifold operations in future work, while retaining the ability to represent nodes in a non-Euclidean space.
>
> Regarding the discussion of societal impact, we agree that it is currently broad and plan to expand it to more thoroughly consider potential biases, for example in applications such as fraud detection.

---

> > ### Author Rebuttal · Reviewer_6419 · 2026-04-05
> >
> > I have no concerns. I have decided to keep my score.

---

> > > ### Author Response · Authors · 2026-04-06
> > >
> > > We sincerely appreciate the reviewer’s feedback and are glad that our work has fully addressed the reviewer’s concerns.

---

### Official Review · Reviewer_hNFj · 2026-03-09

**Soundness:** 3
**Presentation:** 3
**Significance:** 3
**Originality:** 3
**Overall Recommendation:** 4
**Confidence:** 4

**Summary:**

This paper addresses the challenge of Graph Anomaly Detection (GAD) under heterophily, where connected nodes frequently have dissimilar labels.
This paper introduce HSMAD, a novel framework that jointly models heterophily across both spectral and geometric domains.
HSMAD incorporates two key modules: Heterophily-Weighted Spectral Filtering and Geometry-Aware Manifold Updates, which work to generate highly informative node representations. Extensive experiments on six real-world datasets demonstrate the superiority of HSMAD over state-of-the-art baselines.

**Compliance With Llm Reviewing Policy:**

Affirmed.

**Final Justification:**

The author response partially solved my confusion.
However, the discussion regarding sparse graphs in my previous comments remains insufficient.
I will keep the original score.

**Key Questions For Authors:**

See weakness.

**Limitations:**

No.

**Strengths And Weaknesses:**

**Strength**
1. The paper proposes a novel and insightful framework, HSMAD, which integrates heterophily-weighted spectral filtering with curvature-aware manifold routing. It effectively addressed the inherent limitations of conventional Euclidean-based methods that often fail to capture complex structural patterns in heterophilic graphs.

2. This paper is grounded in theoretical analysis.

3. This paper conducted comprehensive experimental validation across six real-world datasets. The detailed ablation studies and sensitivity analysis spectral filter order C and the quantile $\tau$ convincingly demonstrate the contribution of the two modules.

**Weakness**

1. The paper does not provide an analysis of computational complexity or runtime comparison.

2. The paper states that 'homophilic messages are mapped to spherical space, while heterophilic messages are mapped to hyperbolic space.' However, the reviewer questioned why the reverse configuration cannot be used. Please include an ablation study on swapping the assignments or provide relevant theoretical justification.

3. The paper acknowledges that HSMAD performs worse than PMP and CGADM in T-Social. This is attributed to the fact that sparse connections limit availability of informative neighborhood signals. However, the paper did not propose solutions or mitigation strategies for sparse graphs, which are very common in practice.

---

> ### Author Rebuttal · Authors · 2026-03-30
>
> We sincerely thank the reviewer's insightful comments and suggestion to conduct additional experiments.
>
> **W1. Efficiency experiments**
>
> We conducted efficiency comparisons with recent 2025 methods, including DSGAD, CurvGAD, SpaceGNN, and CGADM. On the **Yelp** dataset, HSMAD achieves a favorable trade-off between performance and computational efficiency: it is slightly slower than DSGAD but consistently faster than CurvGAD, SpaceGNN, and CGADM. In terms of memory, HSMAD requires less than CurvGAD. On the large-scale **T-Social** dataset, several baselines (e.g., CurvGAD and SpaceGNN) encounter out-of-memory (OOM) or out-of-time (OOT) issues, whereas HSMAD remains executable, demonstrating its practical scalability.
>
> |Yelp|Training Time (s/epoch)|Inference Time (s/eval)|Train Memory (MB)|Inference Memory (MB)|
> |----|------------------------|------------------------|----------------------|--------------------------|
> |DSGAD|0.0909|0.0060|1193.47|913.70|
> |CurvGAD|4.8877|4.0130|11142.37|3361.36|
> |SpaceGNN|11.0786|2.2711|459.00|334.94|
> |CGADM|2.4930|2.1519|4788.29|4211.10|
> |HSMAD|0.4720|0.2802|5397.17|3386.15|
>
> |T-Social|Training Time (s/epoch)|Inference Time (s/eval)|Train Memory (MB)|Inference Memory (MB)|
> |--------|------------------------|------------------------|----------------------|--------------------------|
> |DSGAD|2.4341|0.0157|23268.58|17741.09|
> |CurvGAD|-|-|OOM|OOM|
> |SpaceGNN|OOT|OOT|-|-|
> |CGADM|13.9265|15.4230|32495.69|20639.04|
> |HSMAD|9.9169|6.4775|65100.89|23613.84|
>
> The additional overhead in HSMAD arises from edge-wise feature modeling and manifold geometry operations, which are specifically designed to capture heterophily and achieve strong performance. These modules inevitably increase computational and memory requirements, but HSMAD remains more resource-efficient than methods such as CurvGAD that also employ manifold representations. We acknowledge this trade-off and plan to further explore resource-efficient strategies in future work to reduce computational and memory costs while retaining performance.
>
> **W2. Ablation Study on swapping the assignments**
>
> Our design is motivated by commonly observed geometric properties. Spherical space (with positive curvature) is generally suitable for modeling locally clustered structures, as it tends to keep representations within a relatively compact region. In contrast, hyperbolic space (with negative curvature) provides greater capacity to represent structurally diverse and distant relationships, which are often associated with heterophilous connections.
>
> To examine this design choice, we conducted an ablation study by swapping the manifold assignments of homophilic and heterophilic messages. We observe that the swapped configuration can achieve comparable or slightly higher F1-macro on some datasets (e.g., Amazon and Weibo), but generally leads to a consistent decrease in AUROC across all six datasets. A possible explanation is that the swapped setting introduces a mild mismatch between structural patterns and geometric properties. Mapping heterophilic messages into spherical space may limit its ability to represent more diverse relationships, while mapping homophilic messages into hyperbolic space may reduce the compactness of locally similar nodes. These effects are not drastic but can influence the model’s discriminative ability, particularly in ranking-based metrics such as AUROC.
>
> Overall, while the swapped configuration can occasionally achieve competitive F1-macro, the original mapping consistently yields better AUROC, supporting our proposed design as a stable and effective choice. We will further explore more adaptive or data-driven mapping strategies in future work.
>
> |Dataset|Amazon| |Yelp| |Weibo| |Tolokers| |T-Finance| |T-Social| |
> |:----:|:----:|:--:|:----:|:--:|:----:|:--:|:----:|:--:|:----:|:--:|:----:|:--:|
> |Metric|F1-macro|AUROC|F1-macro|AUROC|F1-macro|AUROC|F1-macro|AUROC|F1-macro|AUROC|F1-macro|AUROC|
> |HSMAD|0.9283|0.9827|0.8682|0.9447|0.9554|0.9944|0.7275|0.8439|0.9290|0.9751|0.9634|0.9888|
> |Swapped|0.9287|0.9797|0.8395|0.9308|0.9573|0.9924|0.7205|0.8411|0.9251|0.9742|0.9471|0.9817|
>
> **W3. Performance of HSMAD on the T-Social dataset**
>
> As noted, HSMAD performs slightly worse than PMP and CGADM on the T-Social dataset, primarily due to its sparsity, which limits the availability of informative neighborhood signals for both heterophily modeling and message aggregation. We acknowledge that such sparsity is common in real-world graphs and presents a challenging setting for methods relying on structural information. To address this, we plan to investigate mitigation strategies in future work, including neighborhood augmentation, edge enhancement, and structure-aware sampling, which aim to improve robustness under extreme sparsity.

---

> > ### Author Rebuttal · Reviewer_hNFj · 2026-04-07
> >
> > Thank you for detailed response which partially solved my confusion.
> > I will keep the original score.

---

> > > ### Author Response · Authors · 2026-04-07
> > >
> > > We thank the reviewer for the valuable feedback and understand that our response has partially addressed the reviewer’s concerns.
> > >
> > > We would like to clarify that, in our experiments, isolated nodes are handled by adding self-loop edges to enable their participation in message passing and heterophily score training, which serves as a basic mitigation strategy for sparsity. In addition, HSMAD’s design leverages neighborhood aggregation and heterophily modeling, allowing it to exploit available structural information in sparse regions.
> > >
> > > Beyond these measures, we appreciate the reviewer’s suggestions and plan to explore additional strategies in future work, such as generating virtual neighbors or incorporating auxiliary edges, to further improve robustness under extreme sparsity.

---

### Official Review · Reviewer_8AhF · 2026-03-12

**Soundness:** 4
**Presentation:** 3
**Significance:** 3
**Originality:** 3
**Overall Recommendation:** 4
**Confidence:** 3

**Summary:**

This paper addresses the core challenge in Graph Anomaly Detection: the contradiction between edge heterophily and the homophily assumption of GNNs. The authors propose a dual-module framework, HSMAD. The first module, Heterophily-Weighted Spectral Filtering (HWSF), employs a multi-band Beta wavelet filter bank to capture spectral information from both low- and high-heterophily edges. The second module, Heterophily-Routed Manifold Update (HRMU), routes messages to positive/negative curvature manifolds based on predicted heterophily scores for aggregation and update. The model explicitly models heterophily in the spectral domain while performing message passing in more expressive manifold spaces, demonstrating both theoretical depth and engineering feasibility.

**Compliance With Llm Reviewing Policy:**

Affirmed.

**Final Justification:**

The author has fully addressed my concern. I will keep the positive score.

**Key Questions For Authors:**

1. All experiments in this paper are conducted on static, homogeneous graphs. Have the authors considered extending HSMAD to dynamic graph scenarios? If so, which modules would require redesign?
2. Some datasets contain isolated nodes. How are these nodes handled during message passing and heterophily score training? Has consideration been given to generating virtual neighbors or adding auxiliary edges for isolated nodes to enhance their representations?
3. The appendix reveals significant differences in the heterophily ratio distributions of anomalous nodes across datasets. How do these distributional differences impact model performance? Should adaptive strategies be designed for different distribution characteristics?
4. Could the authors provide more details on how critical hyperparameters like manifold curvature k, loss weights, and fusion coefficients are set or learned? For instance, are they learned via backpropagation, or are they preset based on validation performance? What are the initialization strategies?

**Limitations:**

While the appendix briefly mentions model complexity, it lacks a concrete time/space complexity analysis and runtime comparisons with baseline methods. It is strongly recommended that the authors supplement the final version with relevant experiments. This could include training/inference time and memory consumption on graphs of varying scales, as well as scalability evaluations. Such additions would provide a more comprehensive measure of the model's feasibility and efficiency in large-scale, real-world deployment scenarios. For large datasets like T-Social, finer-grained resource analysis would be particularly valuable.

**Strengths And Weaknesses:**

Strengths
1. The paper clearly identifies the conflict between heterophily in GAD and the homophily assumption of GNNs, designing solutions centered on this core issue. The problem formulation is precise.
2. The authors provide three key lemmas—Spectral Bounds under Heterophily Weighting, Stability of the Beta Graph Wavelet Frame, and the First-Order Expansion of k-Addition—with complete mathematical proofs in the appendix. This significantly enhances the credibility of the proposed method.
3. The model jointly models heterophily from both spectral and geometric manifold perspectives. The choice to perform filtering based on the Laplacian matrix rather than the raw adjacency matrix is theoretically sound.
4. Experiments are conducted on six real-world datasets against over a dozen baselines, evaluated across four metrics: F1-Macro, AUROC, AUPRC, and G-Mean. The inclusion of case studies further strengthens the findings.

Weaknesses
1. The performance degradation upon removing either the HWSF or HRMU module individually is relatively small. This may suggest functional overlap or underutilized synergy between the two modules on the current datasets. It is recommended that the authors further analyze their independent contributions under extreme heterophily distributions or design more granular ablation settings (e.g., removing both, alternating usage).
2. The specification of the manifold curvature k is unclear. Details for other critical hyperparameters, such as the loss weights and the fusion coefficients, are also not fully disclosed, affecting reproducibility.the variable s_uv appears in the definition w_uv=exp⁡(-αs_uv) without prior definition; it is presumed to be the predicted heterophily score e ̂_ij.
3. The appendix shows that the node-level heterophily ratio distribution for anomalous nodes in the Weibo dataset is left-skewed, deviating from the overall conclusion that "anomalous nodes exhibit higher heterophily." However, the model still achieves the best performance on this dataset. The authors do not delve into this phenomenon, potentially overlooking important data characteristics.
4. The model introduces multiple learnable weight coefficients and gating mechanisms. Although subsequent experiments suggest a limited impact on performance, this redundancy could pose a risk of overfitting. A parameter sensitivity analysis or an attempt at a simplified design is suggested.

---

> ### Author Rebuttal · Authors · 2026-03-30
>
> We sincerely thank the reviewer's careful analysis and constructive suggestions.
>
> **W1. Additional ablation study**
>
> Removing either HWSF or HRMU individually results in only a moderate performance drop. This relatively small decrease occurs because both modules rely on the same heterophily score, so part of the heterophily signal remains even when one module is removed. Importantly, the two modules are not redundant: HWSF captures spectral (frequency-domain) patterns, while HRMU models geometric relationships, allowing each to enhance representation learning in complementary ways.
>
> To further address the reviewer’s concern, we also evaluated a setting in which the heterophily score is completely removed from both modules (w/o Hete.), leaving only the backbone. In this case, performance drops substantially, particularly on highly heterophilous datasets such as **Yelp, Tolokers, and T-Social**, with consistent degradation across Weibo, Amazon, and T-Finance. These results indicate that explicitly modeling heterophily is the key factor driving the observed performance improvement.
>
> |Dataset|Amazon| |Yelp| |Weibo| |Tolokers| |T-Finance| |T-Social| |
> |:----:|:----:|:--:|:----:|:--:|:----:|:--:|:----:|:--:|:----:|:--:|:----:|:--:|
> |Metric|F1-macro|AUROC|F1-macro|AUROC|F1-macro|AUROC|F1-macro|AUROC|F1-macro|AUROC|F1-macro|AUROC|
> |HSMAD|0.9283|0.9827|0.8682|0.9447|0.9554|0.9944|0.7275|0.8439|0.9290|0.9751|0.9634|0.9888|
> |w/o Hete.|0.9244|0.9819|0.7696|0.8951|0.9428|0.9889|0.6892|0.8149|0.9144|0.9730|0.9313|0.9633|
>
> **W2 and Q4. Clarification of parameters and notation**
>
> In our experiments, the following parameters are learned end-to-end via backpropagation:
>
> - The curvature parameters $\kappa^+$ and $\kappa^-$ are learnable. Each is initialized near zero and mapped through a smooth function to ensure positivity, after which $\kappa^+$ and $\kappa^-$ are assigned positive and negative signs, respectively.
> - Loss weights for nodes and edges are dynamically adjusted during training based on their relative magnitudes, with a small constant added to ensure numerical stability. Each weight is computed to balance the contribution of node and edge losses.
> - Fusion coefficients are learnable parameters passed through a sigmoid function.
>
> Regarding notation, $s_{uv}$ corresponds to the predicted heterophily score $\hat e_{ij}$. We will clarify this in the revised manuscript to ensure consistency and avoid any potential confusion.
>
> **W3 and Q3. Atypical heterophily distribution in the Weibo dataset**
>
> Our analysis indicates that, in most datasets, anomalous nodes exhibit higher heterophily than normal nodes, motivating heterophily modeling and explaining why our model outperforms baseline methods in these cases.
>
> In contrast, in the Weibo dataset, anomalous nodes display similarly low heterophily, consistent with the homophily assumption underlying standard GNNs, which allows baseline GNNs to perform well. Because our model explicitly models heterophily, it excels on datasets with high-heterophily anomalous nodes, while on Weibo, it effectively operates like a conventional GNN, maintaining strong performance despite the low heterophily distribution.
>
> **W4. Potential redundancy and overfitting risk in the learnable weights and gating mechanisms**
>
> Our experiments show that these components do not adversely affect performance. Nevertheless, we agree that they may introduce some overfitting risk, and we plan to explore more lightweight designs in future work to reduce complexity while maintaining flexibility.
>
> **Q1. Extending HSMAD to dynamic graphs**
>
> We acknowledge that the current HSMAD model is developed on static graphs. Extending it to dynamic graph scenarios is an important direction for future research. Our current idea is to jointly model the heterophily of edges along with the temporal information they carry, which would allow the model to capture event-driven incremental updates. Additionally, addressing catastrophic forgetting will be necessary to ensure the model retains knowledge from previous graph snapshots.
>
> **Q2. Handling of isolated nodes**
>
> In our experiments, isolated nodes are handled by adding self-loop edges to allow participation in message passing and heterophily score training. We did not generate virtual neighbors or add auxiliary edges beyond self-loops. In future work, we plan to explore the reviewer's suggested strategies, such as generating virtual neighbors or adding auxiliary edges.
>
> We conducted efficiency comparisons with recent 2025 methods, including DSGAD, CurvGAD, SpaceGNN, and CGADM. Due to space constraints, the results are provided in tables within our responses to other reviewers. These results indicate that our method demonstrates moderate time and memory costs. The primary memory usage stems from the computation and storage of edge features. We acknowledge this limitation and plan to investigate more resource-efficient strategies in future work.

---

> > ### Author Rebuttal · Reviewer_8AhF · 2026-04-03
> >
> > Thanks to the authors' effort on the reply. However, the response to Question 3 isn’t very convincing. Clearly, Weibo, T-Finance, and Tolokers do not fully meet the assumption of right-skewed heteroskedasticity stated in the manuscript; the statement does not hold universally.

---

> > > ### Author Response · Authors · 2026-04-06
> > >
> > > We sincerely thank the reviewer for the careful reading and valuable comments on our manuscript.
> > >
> > > We would like to clarify a potential misunderstanding regarding the term **“distributions shifted to the right”** used in Figure 5. We did not state or assume in the manuscript that anomalous nodes follow a **“right-skewed heteroskedastic”** distribution. This term may be interpreted as suggesting such a pattern, but this is not the intended meaning.
> > >
> > > The phrase **“distributions shifted to the right”** is used only to describe a **relative difference in heterophily levels**: within each dataset, anomalous nodes tend to exhibit higher node-level heterophily than normal nodes. It refers to a shift in **central tendency**, rather than any assumption about skewness, variance structure, or overall distribution shape.
> > >
> > > > “Figure 5. Density distributions of node-level heterophily ratios for normal and anomalous nodes across six datasets. Anomalous nodes exhibit higher heterophily ratios, with distributions shifted to the right.”
> > >
> > > To further clarify this point, we report the mean node-level heterophily ratios across datasets:
> > >
> > > | Dataset         | Amazon | Yelp   | Weibo  | Tolokers | T-Finance | T-Social |
> > > |-----------------|--------|--------|--------|----------|-----------|----------|
> > > | Normal nodes    | 0.0317 | 0.1324 | 0.0228 | 0.3214   | 0.0235    | 0.1003   |
> > > | Anomalous nodes | 0.8975 | 0.8049 | 0.1421 | 0.5238   | 0.4566    | 0.8260   |
> > >
> > > As shown, anomalous nodes exhibit higher mean heterophily than normal nodes across all datasets, including Weibo, T-Finance, and Tolokers.
> > >
> > > We hope this clarification addresses the reviewer’s concern and provides a clearer interpretation of Figure 5. We would greatly appreciate it if our explanation could be taken into consideration when evaluating the manuscript score.

---

### Decision · Program_Chairs · 2026-04-30

**Decision:**

Accept (regular)

**Comment:**

The paper received three reviews and all of them are positive. The reviewers acknowledged that the paper has the following merits:
1. Clearly addresses a key gap in GAD (heterophily vs. homophily assumption)
2. Novel dual-domain architecture (spectral + geometric manifold learning)
3. SOTA performance on six datasets
4. Scales to large graphs

Nevertheless, there are several minor issues:
1. The spectral module is incrementally similar to those used in previous works.
2. The method relies on abundant labeled data (poorly suited for sparse-label scenarios).

In sum, given the relatively novel idea and the impressive performance, I recommend acceptance.